# Potential improvements in global carbon flux estimates from a network of laser heterodyne radiometer measurements of column carbon dioxide

Paul I. Palmer,[1,2] Emily L. Wilson,[3] Geronimo L. Villanueva,[3] Giuliano Liuzzi,[3,4] Liang Feng,[1] A.J. DiGregorio,[3,5] Jianping Mao,[3,6] Lesley Ott,[3] Bryan Duncan[3]

[1]National Centre for Earth Observation, University of Edinburgh, Edinburgh, UK
[2]School of Geosciences, University of Edinburgh, Edinburgh, UK
[3]NASA Goddard Space Flight Center, 8800 Greenbelt Road, Greenbelt, MD 20771, USA
[4]Department of Physics, American University, 4400 Massachusetts Avenue NW, Washington, DC 20016, USA
[5]Science Systems and Applications, Inc., Lanham, MD 20706, USA
[6]Earth System Science Interdisciplinary Center, University of Maryland, College Park, MD 20740, USA

*Correspondence to*: Emily Wilson (Emily.L.Wilson@nasa.gov)

**Abstract.** We present Observing System Simulation Experiments (OSSEs) to evaluate the impact of a proposed network of ground-based miniaturized laser heterodyne radiometer (mini-LHR) instruments that measure atmospheric column-averaged carbon dioxide ($XCO_2$) with a 1 ppm precision. A particular strength of this passive measurement approach is its insensitivity to clouds and aerosols due to its direct sun pointing and narrow field-of-view (0.2 degrees). Developed at NASA Goddard Space Flight Center (GSFC), these portable, low-cost mini-LHR instruments were designed to operate in tandem with the sun photometers used by the AErosol RObotic NETwork (AERONET). This partnership allows us to leverage the existing framework of AERONET's 500+ site global ground network, as well as provide simultaneous measurements of aerosols that are known to be a major source of error in retrievals of $XCO_2$ from passive nadir-viewing satellite observations. We show using the global 3-D GEOS-Chem chemistry transport model that a deployment of 50 mini-LHRs at strategic (but not optimized) AERONET sites significantly improves our knowledge of global and regional land-based $CO_2$ fluxes. This improvement varies seasonally and ranges 58- 81% over southern lands, 47-76% over tropical lands, 71-92% over northern lands, and 64-91% globally. We also show significant added value from combining mini-LHR instruments with the existing ground-based NOAA flask network. Collectively, these data result in improved *a posteriori* $CO_2$ flux estimates on spatial scales of $\sim 10^6$ km$^2$, especially over North America and Europe where the ground-based networks are densest. Our studies suggest that the mini-LHR network could also play a substantive role in reducing carbon flux uncertainty in Arctic and tropical systems by filling in geographical gaps in measurements left by ground-based networks and space-based observations. A realized network would also provide necessary data for the quinquennial global stocktakes that form part of the Paris Agreement.

# 1 Introduction

Two recent satellite instruments have made significant contributions to globally characterizing $XCO_2$: the Japanese Greenhouse gases Observation SATellite (GOSAT) or "IBUKI" launched in 2009 (Kuze et al., 2009), and the Orbiting Carbon Observatory (OCO-2) (Crisp et al., 2017, Eldering et al, 2018) launched in 2014. Both the Fourier Transform
Spectrometer (FTS) in GOSAT and the grating spectrometer in OCO-2 have multiple viewing geometries (nadir, glint, and target) to observe absorption of $XCO_2$, but OCO-2 offers significant improvements in global surface coverage. While GOSAT and OCO-2 have made important advances in observing greenhouse gases from space, any uncharacterized systematic errors can compromise the accuracy of their data (Wunch et al., 2017) and limit the utility of such datasets for inferring surface flux distributions (Basu et al., 2013). Ground-based networks of accurate/precise XCO2 measurements such
as TCCON (Wunch et al., 2017) therefore play an important role in helping to validate these space-borne missions. We describe how we can improve knowledge of the carbon cycle by establishing a network of low-cost, portable mini-LHR (miniaturized laser heterodyne radiometer) instruments that measure $XCO_2$, to fill in gaps left by existing column ground-based networks and space-borne observations. These instruments can be quickly deployed (to be collecting data within a few hours) and can run autonomously in the field with little or no maintenance over a period of months or years.

Ground-based, broad spectral column measurements of $XCO_2$ from the TCCON Fourier Transform Spectrometer (FTS) network have been used to minimize regional systematic errors and serve as a gold standard to validate satellite measurements. In 2010, the TCCON FTS instruments reported an accuracy of ~1 ppm due to bias errors from uncertainties in spectroscopic parameters (Wunch et al., 2010). They resolved this limitation at five of their sites by tying their column-
averaged dry-air mole fractions to the World Meteorological Organization (WMO) *in situ* trace gas measurements scales using aircraft profiles and indicated that they planned to eventually perform similar calibrations at the remainder of their sites. While TCCON products are well characterized, the majority of the 32 TCCON sites are in the Northern Hemisphere, leaving important monitoring gaps in regions where our knowledge of the drivers of carbon cycling is uncertain (Shuur et al., 2008;Commane et al., 2017;Saunois et al., 2016;Le Quéré et al., 2016).

The NASA mini-LHRs are designed to be deployed in tandem with AERONET sun photometers (Holben et al., 1998), taking advantage of their sun-trackers. This partnership provides a pathway to establish a global network of mini-LHRs by leveraging AERONET's 500+ site network and offers a simultaneous measure of aerosol optical depth (AOD) that is a necessary input for satellite retrievals (Butz et al, 2009). Similar to TCCON, mini-LHRs can collect data during breaks in
cloud coverage thereby offering the potential for new data products in formerly underrepresented regions such as the Amazon basin, southern Asia monsoon areas, and the Arctic. These vulnerable geographic regions are not well covered by OCO-2 and GOSAT. Here, using numerical experiments, we simulate a strategic (but not optimized) deployment of 50 mini-

LHR instruments to AERONET sites and evaluate how this increase in measurement density impacts knowledge of regional and global carbon fluxes.

## 2 Mini-LHR Instrument Configuration

The mini-LHR is a ground-based, passive, sun-viewing instrument that observes trace gases in the atmospheric column. It has been under development at NASA Goddard Space Flight Center (GSFC) since 2009 (Melroy et al., 2015;Clarke et al., 2014;Wilson et al., 2014;Wilson and McLinden, Filed 2012, Issued 2014) and while earlier versions exclusively measured $XCO_2$, the current version observes both $XCO_2$ and $XCH_4$. Current challenges associated with our understanding emissions of $CH_4$ (Wolf et al., 2017) will result in a different network design. The mini-LHR has been tested at altitudes ranging from sea level to 3,400 meters, and in climates that include tropical, subtropical, and temperate zones, extending to just below the Arctic Circle and has shown consistent precisions of 1 ppm $XCO_2$ and 10 ppb $XCH_4$ for hourly data products. Fig. 1 shows a mini-LHR monitoring $XCO_2$ and $XCH_4$ over thawing permafrost at a remote site in the Bonanza Creek Research Forest near Fairbanks, Alaska. The goal of these field tests was to both improve the quality of the data product as well as test the durability of commercial components that were intended for indoor lab use.

The mini-LHR measures $XCO_2$ by scanning the $CO_2$ absorption feature near 1.61 μm. Fig. 2 shows the current configuration of the system and Table 1 lists key system parameters. Sunlight is collected with a fibre-coupled, 0.2-degree field-of-view collimator that is non-invasively connected to an AERONET sun tracker. Once collected, sunlight is modulated with a fibre switch, superimposed with infrared laser light from a distributive feedback laser in a single mode fibre coupler, and then mixed in a fast photoreceiver/InGaAs detector to produce an RF beat signal. The RF receiver separates RF and DC outputs, and the RF signal is amplified, filtered, and then detected with a square-law detector. The resulting signal is measured with a lock-in amplifier referenced to the fibre switch frequency as the laser scans across an absorption feature. A microprocessor controls the laser scanning and data collection. The mini-LHR has spectral sampling resolution of $\sim 0.013$ cm$^{-1}$ which is 15 times higher than GOSAT ($\sim 0.2$ cm$^{-1}$), 20 times higher than OCO-2 ($\sim 0.3$ cm$^{-1}$) and slightly higher than TCCON ($\sim 0.02$ cm$^{-1}$). Individual scans of the $CO_2$ feature are collected at 2-minute intervals throughout the day during sunlight hours when clouds are not present and averaged into hourly data products.

## 3 Data Processing and Retrieval

Averaged absorption scans are analyzed to extract column mole fractions of $CO_2$ using custom analysis software developed at GSFC that is similar to the approach used by TCCON. There are two main steps involved in processing data: (1) simulating the spectra (mathematically simulating what the mini-LHR observes in the atmosphere), and (2) fitting the simulation to the data to extract the abundance of $XCO_2$.

We simulate the spectra using the Planetary Spectrum Generator (PSG), which is an online tool developed at NASA GSFC (Villanueva et al., 2016;Villanueva et al., 2015) for synthesizing Earth and planetary spectra (atmospheres and surfaces) for a broad range of wavelengths (0.1 μm to 100 mm, UV/Vis/near-IR/IR/far-IR/THz/sub-mm/Radio) from any observatory, orbiter, or lander. This is achieved by combining several state-of-the-art radiative transfer models, spectroscopic databases and planetary databases. The PSG code includes refraction of sunlight through the atmosphere as well as a computationally efficient scattering package that incorporates the latest radiative transfer numerical methods (Villanueva et al., 2015;Smith et al., 2009), and is parameterized for LTE (Local-Thermodynamic Equilibrium) calculations. While scattering is not required for direct sun-viewing measurements, the scattering package contains a treatment of aerosols; the extinction portion of which is needed to properly model the continuum shape. The PSG is operated remotely by employing a versatile online Application Program Interface (API). The API operates by sending a configuration file to the PSG servers. Upon reception of the configuration file, PSG computes and returns the spectra.

Part of this simulation includes the Modern-Era Retrospective analysis for Research and Applications, Version 2 (MERRA-2) data set which provides meteorological inputs (Reichle et al., 2011;Rienecker, 2011) and provides a 72-layer model of the atmosphere. The retrieval employs MERRA-2 database to define the state and *a-priori* values for the atmosphere. We "perturb" the $CO_2$ profile by a scaler, which is the value that it is actually being retrieved by the retrieval algorithm. MERRA is the Modern-Era Retrospective Analysis for Research and Applications database, which is the latest atmospheric reanalysis of the modern satellite era produced by NASA's Global Modeling and Assimilation Office, which incorporates information from hundreds of orbiters and ground stations since 1980 and provides global three-dimensional of atmospheric parameters (e.g., temperature, abundance profiles, aerosols). Specifically, our retrieval works with the M2I3NVASM component, which provides assimilated meteorological fields (pressure, temperature, water vapor, ozone, and water ice clouds) from the surface to ~80 km (72 layers) with a cadence of 180 minutes, and spatial resolution of ~0.5 degrees (576 x 361). The values are further refined temporally and spatially to a resolution of better than 1 km employing the USGS-GTOPO30 topographic maps and considering a hydrostatic equilibrated atmosphere within every bin. Our code computes temperature (T) and pressure (P) abundances for Earth by first selecting a set of 6 standard profiles based on season and latitude: Tropical', 'Midlatitude-Summer', 'Midlatitude-Winter', 'Subarctic-Summer', 'Subarctic-Winter', 'US-Standard' (Anderson et al., 1986). These profiles provide abundances for a myriad of species and basic temperature and pressure profiles. The code then extracts P, T, $O_3$, $H_2O$ and water ice abundances from the MERRA-2 database for this location and time. The MERRA-2 grid is described on a coarse grid and it does not contain fine elevation information and therefore the GTOPO30 topography database (~1 km resolution) is also used to derive the exact elevation of the mini-LHR site location. The information from MERRA-2 at a particular geolocation is then refined in elevation, e.g. using scale heights, using this high-resolution topographic map.

Our code generates an initial configuration file that establishes the location and date/time of the measurement. Using this configuration file, the code calls the PSG/API and this returns all of the geometry parameters (air mass, phase angle, etc.) and an *a priori* vertical profile based on the date and location. Then, using this configuration file, the program goes into the fitting routine that calls the PSG/API to calculate spectra by fitting the $CO_2$ abundance using an optimal estimation approach.

The fit perturbs the $CO_2$ abundance and obtains a fit based on the Levenberg-Marquardt algorithm which is an iterative least-squares curve fitting procedure.

## 3.1 Calibration and Validation of mini-LHR Data

Mini-LHR instruments periodically undergo a calibration/validation procedure at NASA GSFC to track performance and establish documented traceability of column data products. In particular, we calculate and report measurement precision,

measurement error, and measurement bias, as defined by the Vocabulaire International de Metrologie (VIM) (Meaures, 2012).

We estimate measurement precision (standard deviation) by routine laboratory calibrations. In the calibration procedure, the mini-LHR instrument scans a NIST (National Institute of Standards and Technology) traceable atmospheric mixture of gases

(NIST Traceable Reference Material Program for Gas Standards, 2016) in a 36-meter Herriot absorption cell. The NIST traceable atmospheric mixture of gases fulfils the criteria of a measurement standard with a negligible measurement uncertainty. The calibration gas standards are specified in the Code of Federal Regulations (CFR) to calibrate instruments used to monitor regulated emissions. The standard deviation of the points in the scan provides an estimate of precision while regular calibrations track any measurement bias (systematic measurement error).

To assess accuracy of the mini-LHR during its development, two short-duration side-by-side comparisons were completed between the mini-LHR and TCCON stations at Park Falls, Wisconsin in 2012 and at Caltech in Pasadena, CA in 2014. Sample scans from these comparisons are shown in Fig. 3 with fits from the PSG retrieval. Data from 2014 showed a significant improvement in agreement with the TCCON $CO_2$ measurements because it was possible to collect more scans

within a shorter timeframe: The 2012 data is the average of three scans collected over the period of one hour and the 2014 data is the average of five scans collected over the period of a half hour. A longer-duration side-by-side comparison is planned with the TCCON located at NASA Armstrong Flight Research Center (AFRC). In addition to laboratory calibrations, potential bias between mini-LHR instruments will also be addressed by regularly comparing a "standard" mini-LHR that is co-located at the NASA/AFRC TCCON with all other mini-LHR instruments. TCCON FTS instruments

measure column $CH_4$ and $CO_2$ at the same wavelengths but at lower resolution than the mini-LHR. While TCCON has a well-documented history of characterization, we refer to this as an "estimate" of measurement error due to differences in resolution and because there are known biases between TCCON sites (1% for $CO_2$ at US sites and 1.1±0.2% at European sites) (Wunch et al., 2010).

For passive satellite observations, scattering from clouds and aerosols are known to be a significant source of retrieval error for $XCO_2$ (Mao and Kawa, 2004;Aben et al., 2007;Uchino et al., 2012;Yoshida et al., 2013). This is primarily because these are nadir-pointing instruments that view sunlight reflected on a portion of ground that is illuminated by direct sunlight as well as scattered sunlight from clouds and aerosols. In contrast, ground-passive measurements have a narrow field-of-view (FOV) and point directly at the sun.  The TCCON at Park Falls, WI for example, has a FOV of ~0.14 degrees and mini-LHRs have a FOV of ~0.2 degrees (compared to the sun which has a field of view of ~0.5 degrees).  Because the FOVs of these instruments are narrower than that of the sun, their light collection optics do not accept the scattered light outside of this FOV.  Consequently, the mini-LHR and TCCON are mainly impacted by extinction, resulting in lower levels of sunlight reaching these ground instruments and lower signal-to-noise levels. Solar intensity variations that impact TCCON are corrected by dividing the interferograms by the unmodulated DC signal (Keppel-Aleks et al, 2007).  For the mini-LHR, transmittance scans are corrected for extinction by dividing scans by a fitted baseline that tracks fluctuations in solar irradiance.

## 4 Theoretical Potential of Network to Improve Knowledge of Regional Carbon Fluxes

We use numerical experiments to provide an upper limit on the theoretical potential of the proposed network on reducing the uncertainty of regional carbon flux estimates. The approach we take is to define a closed-loop experiment, often called Observing System Simulation Experiments (OSSEs), in which we define the true atmospheric state using a global 3-D model of atmospheric chemistry and transport driven by true surface fluxes. This true atmospheric state is then sampled, as it would be by the mini-LHRs (e.g. time, location and vertical sensitivity). We then generate a complementary set of model values that are generated from an independent surface flux inventory, including differences in the magnitude and distribution of fluxes; we use this independent inventory as our *a priori* for the OSSEs. We infer the *a posteriori* fluxes from measurements using an ensemble Kalman Filter.

We use v9.02 of the GEOS-Chem model of atmospheric chemistry and transport (http://geos-chem.org) driven by GEOS-5 analysed meteorological fields that includes a simulation of atmospheric $CO_2$ that has been evaluated with a range of ground-based, aircraft and satellite observations (Feng et al., 2009; Feng et al., 2011; Feng et al., 2016). For our experiments, we use the model at a horizontal resolution of 4º latitude x 5 º longitude for an arbitrary year, which in our experiments is 2014. We use monthly ODIAC fossil fuel emissions (Oda and Maksyutov, 2011), monthly ocean biosphere fluxes (Takahashi et al., 2009), and weekly biomass burning emissions from GFEDv3 (van der Werf et al., 2010).

Currently, it is common practice to assume that most of the uncertainty in atmospheric $CO_2$ stems from natural fluxes so other sources are typically assumed to be well described by existing inventories (e.g. Gurney et al, 2008). While this practice

is slowly being challenged by the community, we retain these assumptions for the purpose of our theoretical calculations. To define our true atmospheric state we use three-hourly land biosphere fluxes from the ORCHIDEE land surface model (Krinner et al, 2005) and in a separate model calculation to define our *a priori* state we use three-hourly land biosphere fluxes from CASA (Olsen and Randerson, 2004). Fig. 4 shows there are significant seasonal differences in the magnitudes and distributions of ORCHIDEE and CASA land biosphere $CO_2$ fluxes so that our OSSE provides a rigorous test of the theoretical data.

For the purposes of inter-comparability of impacts of different data, we have ignored any source of systematic error in the different measurements or the transport model that links *a priori* information to 4-D atmospheric mole fractions of $CO_2$. Even sub-ppm levels of uncharacterized systematic error in atmospheric measurements will significantly compromise our ability to infer unbiased regional $CO_2$ flux estimates.

Our retrieval simulation requires a matrix of averaging kernels ($\mathbf{A}$) that describes the sensitivity of the retrieved state vector $\hat{x}$ (in this case, a vector describing the vertical profile of $CO_2$ described over *n* atmospheric layers) to the "true" state vector $x$, for different values of solar zenith angle throughout the day. Using the standard convention, upper case and lower case emboldened variables denote a matrix and vector, respectively, and superscripts -1 and T denote matrix inverse and transpose operations, respectively. The averaging kernel is calculated as (Rodgers, 2000;Liuzzi et al., 2016):

$$\mathbf{A} = \frac{\partial \hat{x}}{\partial x} = (\mathbf{S}_a^{-1} + \mathbf{K}^{\mathrm{T}}\mathbf{S}_\epsilon^{-1}\mathbf{K})^{-1}\mathbf{K}^{\mathrm{T}}\mathbf{S}_\epsilon^{-1}\mathbf{K}, \tag{1}$$

where $\mathbf{S}_a$ is the *a priori* error covariance matrix (size *n* x *n*); $\mathbf{S}_\epsilon$ is the measurement error covariance matrix (size *m* x *m*, where *m* denotes the length of the radiance vector); and $\mathbf{K}$ is the matrix of weighting functions describes the derivative of the radiance with respect to a change in the $CO_2$ profile (size *m* x *n*). The matrix of averaging kernel is used here to describe the instrument sensitivity to changes in $CO_2$ so we can simulate mini-LHR $XCO_2$ column measurements in GEOS-Chem. For simplicity, we assume $\mathbf{S}_a$ and $\mathbf{S}_\epsilon$ are diagonal and represent the square of the background variability of $CO_2$ concentration in each atmospheric layer and the square of the instrument noise, respectively.

The sum of the rows of $\mathbf{A}$ corresponds to summing the retrieval sensitivities to the $CO_2$ in each atmosphere layer, and describe the sensitivity of the atmospheric column to a change in atmospheric $CO_2$ in the vertical profile, i.e. the column averaging kernel $\mathbf{a}$. In the ideal case, each summed row would be close to unity. Fig. 5 shows that for a variety of solar zenith angles, a > 0.8 in the troposphere but falls off quickly in the stratosphere, consistent with TCCON averaging kernels (Wunch et al., 2011).

We also calculate the number of Degrees of Freedom (DOF) of the retrieval as the trace of **A**, which estimates the number of independent pieces of information that can be derived from retrieval. In the case of the column $XCO_2$ retrieval, this should be close to or higher than 1; values less than one indicate the influence of *a priori* information (Camy-Peyret et al., 2017). We find that with a SNR value of 500 and an assumed *a priori* variance of 5% for $CO_2$ concentrations results in a DOF between 0.88 and 2.10.

To infer regional fluxes of $CO_2$ from the measurements, we use an established ensemble Kalman Filter approach (Feng et al., 2009;Feng et al., 2016;Feng et al., 2017). For brevity, we refer the reader to Feng et al 2009 for a detailed description of the approach and its application within GEOS-Chem. We adopt a uniform 50% *a priori* uncertainty and assume a conservative 1.5 ppm for individual measurement and model transport errors. To characterize the impact of the mini-LHR measurements on the *a priori* knowledge we use a metric that describes how uncertainty of fluxes have reduced after the *a priori* has been informed by the measurements: $\gamma = 1 - \mathbf{S}_c^{ii}/\mathbf{S}_d^{ii}$, where $\mathbf{S}_c^{ii}$ and $\mathbf{S}_d^{ii}$ denote the diagonal elements of the *a posteriori* and *a priori* $CO_2$ flux error covariance matrices, respectively. A larger value for $\gamma$ denotes a larger scientific impact of the observations. We also report comparisons between true, *a priori*, and *a posteriori* $CO_2$ fluxes over key geographical regions. Together, our use of two independent land biosphere flux inventories, the $\gamma$ metric and the inter-comparison of fluxes provide a transparent theoretical assessment of the LHR data to quantify geographical fluxes of $CO_2$.

We performed three experiments to estimate true $CO_2$ fluxes using: 1) TCCON measurements with their current measurement configuration (Fig. 4); 2) mini-LHR measurements collected at an enhanced number (50) of sites ; and 3) the combined data sets from mini-LHR measurements and selected surface flask sites to study the added value of mini-LHR data to the existing NOAA Earth System Research Laboratory ground-based network of mole fraction data that is commonly used to infer regional $CO_2$ fluxes (e.g. Peylin et al, 2013). We conservatively assume a mini-LHR measurement precision of 1.5 ppm in our experiments, however, the current precisions for the mini-LHR and TCCON data products are <1 ppm for one-hour data products (Wunch et al., 2010;Messerschmidt et al., 2011;Wilson et al., 2017;Melroy et al., 2015). Instrument biases were not included in these OSSE runs because our focus is the relative performance of different ground-based remote sensing networks.

Table 2 lists the proposed enhanced distribution of mini-LHR instruments. These sites were initially chosen to target regions where AERONET sites already existed and where there are gaps in the existing *in situ* measurements, TCCON measurements, and satellite observations. Consideration was also given to accessibility, acknowledging evolving political environments. Consequently, the enhanced network has not been optimized to minimize carbon flux uncertainties.

Table 3 reports the TCCON sites used in our numerical experiments. We simulate TCCON XCO2 observations using the same approach as we use for the mini-LHR instruments. For TCCON, we use $CO_2$ averaging kernels from the latest TCCON XCO2 retrievals of version GCC2014 (Wunch et al., 2011).

Table 4 shows the surface flask sites used in our joint assimilation experiment. They are a subset of the NOAA ground-based network, which are chosen mainly based on their data continuity in recent years (i.e., 2009-2016). In our experiments, we use the real availability of the flask data in the compiled surface data set (GLOBALVIEW-v3.2), while simulating the observation values by sampling model surface $CO_2$ concentrations at the observation location and time.

Figure 6 compares the annual mean deviation between true (ORCHIDEE) and *a posteriori* fluxes inferred from the TCCON, the enhanced mini-LHR network, and the mini-LHR+NOAA flask observations. We acknowledge that the primary purpose of TCCON is to provide a ground-truth for satellite observations and are not optimized for surface flux estimation. Nevertheless, we find that the current TCCON network can generally reduce the systematic bias between the *a priori* and the true state, particularly over northern midlatitude land region that reflects observation coverage. As expected, the uneven and
coarse coverage over tropical and southern land regions results in a large-scale dipole effect between tropical south America and other tropical and extratropical regions, e.g. Australia. Sensitivity experiments (not shown) show that the performance of the mini-LHR instruments distributed using the current TCCON measurement configuration is comparable with the TCCON instruments, despite larger assumed random errors. We find this is primarily due to the comparable role of instrument and atmospheric transport model errors (1.5 ppm), particularly at the 4º latitude x 5º longitude horizontal
resolution employed here. Using finer-scale meteorology could very well reduce model error but knowledge of this error is poorly defined with no robust quantitative method currently available.

*A posteriori* fluxes inferred from the enhanced mini-LHR network significantly improves agreement with true fluxes compared to TCCON measurement configuration, particularly over tropical land regions. Annual mean deviations are
generally smaller than $2\times10^{13}$ molec/cm$^2$/s and vary on the grid scale throughout the year for many continental regions. We find that including the NOAA flask data helps to reduce these variations, in particular over North America and Europe where the coverage by the surface data is densest (Figure 6). Figure 7 compares the root-mean-square-error (RMSE) for our three inversion experiments relative to the true state. First, all our inversions show much smaller deviations compared to the *a priori* values. The enhanced mini-LHR network performs significantly better over tropical lands such as tropical South
America and tropical North Africa.

Figs. 8 and 9 summarize the agreement between *a priori*, true, and *a posteriori* $CO_2$ fluxes from our three inversions for hemispheric-scale land regions. TCCON broadly reproduces true fluxes, but the current TCCON configuration is insensitive to some geographical regions (e.g. Tropical South America and North Africa), as expected. Fluxes inferred from data from

the mini-LHR network , independently and in combination with NOAA flask data, are closer to the true state over most parts of the world, e.g. North Land summer months, as expected.  Fig. 8 and 9 also show that on the large spatial scales we have studied, the NOAA flask data provide a modest amount of additional information to the mini-LHR network. This suggests that a ground-based remote sensing network that provides calibrated, high-frequency observations of column $CO_2$ has

comparable performance with the existing *in situ* network on larger continental spatial scales.

Fig. 10 shows that the mini-LHR enhanced network of 50 sites results in global and significant improvements in our knowledge of $CO_2$ fluxes. Significant values of the error reduction $\gamma$ are found over most of North America and Eurasia  as well as over South America and central and southern Africa. There is a similar geographical distribution of improvements

during boreal summer months, but with larger values over North America and Eurasia including the northernmost latitudes. Similar calculations for the TCCON network show comparable levels of improvement but are more spatially limited, particularly over the northern hemisphere. We show there is clear value in combining *in situ* flask data with the mini-LHR network, with significant improvements in $CO_2$ fluxes particularly over North American and Eurasia.

The mobility of mini-LHR sensors, allows us to locate them in remote environments where an AERONET site is already established.  This includes, in particular, tropical ecosystems where the physical environment is challenging for large-scale instrument installations, and at polar latitudes where space-borne measurements are compromised because of low solar illumination and low surface reflectance over snow/ice. This suggests that the mini-LHR network could play a substantive role in an Arctic monitoring network, particularly during spring and autumn months.

Our results demonstrate the complementarity of the mini-LHR and *in situ* flask networks. Calibrating sensors from the TCCON and mini-LHR networks represent additional observational constraints on $CO_2$ fluxes that rival knowledge inferred from the current *in situ* observation network over large-scale geographical regions (*e.g.*, North America and Eurasia), and outperform for other regions (*e.g.*, tropics). The *in situ* networks provide an invaluable record on the changing carbon cycle

by putting present-day changes in an historical context, whose value is reduced if they are terminated.

**5 Concluding Remarks**

The development of the mini-LHR technology is ongoing but this computational study already builds on a growing body of work that has characterized its error budgets and its in-field performance (Wilson et al, 2013; Clarke et al, 2014; Melroy et al, 2015).

With a modest deployment of mini-LHR instruments to 50 sites, numerical experiments with the GEOS-Chem model indicate that the resulting $XCO_2$ data products lead to improvements of carbon flux uncertainties ranging from 58% to 81%

over southern lands, 47% to 76% over tropical lands, 71% to 92% over northern lands, and 64% to 91% globally. Because mini-LHRs leverage AERONET's global network of more than 500 sites worldwide, additional instruments can be rapidly added to target specific areas of uncertainty such as thawing permafrost emissions in the Arctic or tropical ecosystems in the mid-latitudes.  In addition to infrastructure, co-location of these instruments provides a simultaneous measurement of aerosol

optical depth which is necessary to evaluate and correct aerosol scattering effects in $XCO_2$ satellite retrievals and consequent uncertainty in local and regional carbon flux estimates. Sun-viewing mini-LHR instruments are not impacted by some of the issues that degrade the quality of airborne or space-borne techniques that use reflected sunlight: surface reflectivity (e.g., darkness and angular dependence), surface roughness (sunlight path-length), geo-location error, and aerosol /cloud scattering. Together with the capability of measuring through gaps in cloud cover and continuous observation during

daylight hours, the mini-LHR surface network in tandem operation with AERONET could provide full global and seasonal observation coverage and offer a necessary validation product for orbital missions.

However, the modelling study is largely agnostic to the underlying technology. Consequently, a similar result would be obtained, for example, by using a network of Bruker EM27-Sun instruments after they have been modified to withstand

inclement weather and can run exclusively on solar power. A growing network of inter-calibrated ground-based remote sensing units, as part of a global carbon measurement system, must strike a balance between a diffuse network of gold-standard spectrometers (TCCON), a larger network of intermediate (cost/performance) spectrometers (COCCON), with an even larger network of cheaper, less precise autonomous spectrometers that can be deployed in high-risk/high-reward environments where we remain data poor (e.g. tropics).

**Acknowledgements**

Development of the mini-LHR was supported by the NASA/GSFC Internal Research and Development (IRAD) program, the NASA/GSFC Science Innovation Fund (SIF), and the NASA Interdisciplinary Science (IDS) Program (NNH127DA001N). Work at the University of Edinburgh was partly funded by the NERC National Centre for Earth Observation (NCEO). P. I. Palmer gratefully acknowledges funding from his Royal Society Wolfson Research Merit Award. We would also like to

25 thank Brent Holben and the AERONET team for their ongoing support, Laura Iraci and Jim Podolske from NASA Ames Research Centre (ARC) for support with TCCON access and data at NASA/AFRC.

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

**Table 1:** Parameters used to calculate mini-LHR averaging kernels for the Bonanza Creek site. The molecules analyzed in the 72 layer atmosphere included $H_2O$, $CO_2$, $O_3$, $N_2O$, $CO$, $CH_4$, $O_2$, $N_2$ and the level of variability was 0.05.

| Parameter | Value | Units |
|---|---|---|
| Instrument lower wavelength | 1.61137 | μm |
| Instrument upper wavelength | 1.641165 | μm |
| Instrument resolution at FWHM | 0.000003 | μm |
| Instrument's data SNR | 500 | d |
| a-priori variance | 5 | % |
| Zenith angle range | 0-60 | deg |
| Latitude | 64 + 42.055/60.0 | N degrees |
| Longitude | 360 - (148 + 18.763/60.0) | E degrees |
| UT date/time of observations to extract MERRA values | '2016/05/30 19:15' | yyyy/mm/dd hh:mm |

**Table 2:** List of 50 selected AERONET sites with mini-LHR installed in the future for the OSSE study.

| Site | Latitude | Longitude | Site | Latitude | Longitude |
|---|---|---|---|---|---|
| Amsterdam_Island | -37.8 | 77.6 | NASA/GSFC, MD, USA | 39.0 | -76.9 |
| Arica, Chile | -18.5 | -70.3 | ND_Marbel_Univ, Philippines | 6.5 | 124.8 |
| Birdsville, Australia | -25.9 | 139.3 | NEON-Disney, CO, USA | 28.0 | -81.4 |
| Bribane, Australia | -27.5 | 153.0 | Nha Trang, Vietnam | 12.2 | 109.2 |
| Cairo, Egypt | 30.1 | 31.2 | Omkoi, Thailand | 17.8 | 98.4 |
| CEILAP-BA, Argentina | -34.6 | -58.5 | Park Falls, WI, USA | 45.9 | -90.3 |
| Churchhill, Canada | 58.7 | -93.8 | Penn State, PA, USA | 40.8 | -77.9 |
| Cuiaba, Brazil | -15.6 | -56.1 | Pontianak, Indonesia | 0.1 | 109.2 |
| Dhaka, Bangladesh | 23.7 | 90.4 | Pretoria, South Africa | 25.8 | 28.2 |
| Edinburgh, UK | 55.9 | -3.2 | Pune, India | 18.5 | 73.8 |
| Fairbanks, Alaska, USA | 64.8 | -147.7 | Red Mountain Pass, CO, USA | 37.9 | -107.7 |
| Gobabeb, Namibia | -23.6 | 15.0 | Rio_de_Janeiro_UFRJ, Brazil | -22.8 | -43.3 |
| Ittoqqortoormiit, Greenland | 70.5 | -21.6 | Rio Franco, Brazil | 10.0 | -67.9 |
| Irkutsk, Russia | 51.8 | 103.1 | Santiago, Chile | -33.5 | -70.6 |
| Kaiping, China | 22.4 | 112.7 | Sao Paulo, Brazil | -23.6 | -46.6 |
| Kelowna, Canada | 49.9 | -119.4 | SEGC, Africa | -0.2 | 11.6 |
| Kibale, Uganda | 0.5 | 30.4 | South_Pole, Antarctica | -90.0 | 77.3 |
| Lake_lefroy, Australia | -31.3 | 121.7 | Tamanrasset, Algeria | 22.8 | 5.5 |
| Lanzhou, China | 36.0 | 103.0 | Taylor_Ranch_TWRS, ID, USA | 45.0 | -114.8 |
| Manaus, Brazil | -3.2 | -60.0 | Tomsk, Russia | 56.5 | 85.0 |
| Mauna Loa, HI, USA | 19.5 | -155.6 | Ussuriysk, Russia | 43.7 | 132.2 |
| Mexico City, Mexico | 19.0 | -99.1 | WITS, South Africa | -26.2 | 28.0 |
| Monterey, Canada | 36.6 | -121.9 | Yakutsk, Russia | 62.0 | 129.7 |
| NASA/AFRC, CA, USA | 34.6 | -118.1 | Yekaterinburg, Russia | 57.0 | 59.5 |
| NASA/ARC, CA, USA | 37.4 | -122.1 | Yellowknife, Canada | 62.4 | -114.4 |

**Table 3:** TCCON stations used in this OSSE study.

| Site | Latitude [deg] | Longitude [deg] | Site | Latitude [deg] | Longitude [deg] |
|---|---|---|---|---|---|
| Arrival Heights, Antarctica | -77.8 | 166.7 | Karlsruhe, Germany | 49.1 | 8.4 |
| Anmyeondo, Korea | 36.5 | 126.3 | Lamont, OK, USA | 36.6 | -97.5 |
| Ascension Island | -7.9 | -14.3 | Lauder, New Zealand | -45.0 | 169.7 |
| Bialystok, Poland | 53.2 | 23.0 | Los Alamos, NM, USA | 35.87 | -106.32 |
| Bremen, Germany | 53.1 | 8.9 | Ny Alesund, Spitsbergen | 78.9 | 11.9 |
| Burgos, Philippines | 18.5 | 120.7 | Orleans, France | 48.0 | 2.1 |
| Caltech, USA | 34.1 | -118.1 | Paris, France | 48.8 | 2.4 |
| Darwin, Australia | -12.4 | 130.9 | Park Falls, WI, USA | 45.9 | -90.3 |
| Darwin, Australia | -12.5 | 130.9 | Reunion Island | -20.9 | 55.5 |
| Dryden, USA | 35.0 | -117.9 | Rikubetsu, Japan | 43.5 | 143.8 |
| East Trout Lake, Canada | 54.4 | -105.0 | Saga, Japan | 33.2 | 130.3 |
| Eureka, Canada | 80.1 | -86.4 | Sodankyla, Finland | 67.4 | 26.6 |
| Garmisch, Germany | 47.5 | 11.1 | Tsukuba, Japan | 36.1 | 140.1 |
| Harwell, Oxfordshire | 51.6 | -1.32 | Wollongong, Australia | -34.4 | 150.9 |
| Hefei, China | 31.90 | 118.67 | Yekaterinburg, Russia | 57.04 | 59.55 |
| Izana, Tenerife | 28.3 | -16.5 | Zugspitze, Germany | 47.4 | 11.0 |

**Table 4:** NOAA Flask sites used in the OSSE experiments.

| Flask Site | Latitude [deg] | Longitude [deg] | Flask Site | Latitude [deg] | Longitude [deg] |
|---|---|---|---|---|---|
| ABP | -12.77 | -38.17 | LLB | 54.95 | -112.45 |
| ALT | 82.45 | -62.51 | LLN | 23.47 | 120.87 |
| AMS | -37.80 | 77.54 | LMP | 35.52 | 12.62 |
| AMY | 36.54 | 126.33 | MAA | -67.62 | 62.87 |
| ARA | -23.86 | 148.48 | MBC | 76.25 | -119.35 |
| ASC | -7.97 | -14.40 | MEX | 18.98 | -97.31 |
| ASK | 23.26 | 5.63 | MHD | 53.33 | -9.90 |
| AVI | 17.75 | -64.75 | MID | 28.21 | -177.38 |
| AZR | 38.77 | -27.38 | MKN | -0.06 | 37.30 |
| BAL | 55.35 | 17.22 | MLO | 19.54 | -155.58 |
| BCS | 23.30 | -110.20 | MQA | -54.48 | 158.97 |
| BGU | 41.97 | 3.23 | NAT | -5.80 | -35.19 |
| BHD | -41.41 | 174.87 | NMB | -23.58 | 15.03 |
| BKT | -0.20 | 100.32 | NWR | 40.05 | -105.59 |
| BME | 32.37 | -64.65 | OBN | 55.11 | 36.60 |
| BMW | 32.27 | -64.88 | OPW | 48.30 | -124.63 |
| BRW | 71.32 | -156.61 | OTA | -38.52 | 142.82 |
| BSC | 44.18 | 28.67 | OXK | 50.03 | 11.81 |
| CBA | 55.21 | -162.72 | PAL | 67.97 | 24.12 |
| CFA | -19.28 | 147.06 | PDM | 42.94 | 0.14 |
| CGO | -40.68 | 144.69 | PSA | -64.92 | -64.00 |
| CHR | 1.70 | -157.15 | PSA | -64.92 | -64.00 |
| CIB | 41.81 | -4.93 | PTA | 38.96 | -123.74 |
| CMO | 45.48 | -123.97 | RK1 | -29.20 | -177.90 |
| CPA | -12.42 | 130.57 | RPB | 13.17 | -59.43 |
| CPT | -34.35 | 18.49 | SDZ | 40.65 | 117.12 |
| CRI | 15.08 | 73.83 | SEY | -4.68 | 55.53 |
| CRZ | -46.43 | 51.85 | SGI | -54.00 | -38.05 |
| CYA | -66.28 | 110.52 | SGP | 36.61 | -97.49 |
| DRP | -59.00 | -64.69 | SHM | 52.71 | 174.13 |
| DSI | 20.70 | 116.73 | SIS | 60.90 | -1.26 |
| EIC | -27.16 | -109.43 | SMO | -14.25 | -170.56 |
| ELL | 42.58 | 0.96 | SPO | -89.98 | -24.80 |
| ESP | 49.38 | -126.54 | STC | 54.00 | -35.00 |
| FKL | 35.34 | 25.67 | STM | 66.00 | 2.00 |
| GMI | 13.39 | 144.66 | STP | 50.00 | -145.00 |
| GOZ | 36.05 | 14.89 | SUM | 72.60 | -38.42 |
| GPA | -12.25 | 131.05 | SYO | -69.01 | 39.59 |
| HBA | -75.62 | -26.21 | TAC | 52.52 | 1.14 |
| HPB | 47.80 | 11.02 | TAP | 36.74 | 126.13 |
| HSU | 41.06 | -124.75 | THD | 41.05 | -124.15 |
| HUN | 46.95 | 16.65 | TIK | 71.60 | 128.89 |
| ICE | 63.40 | -20.29 | USH | -54.85 | -68.31 |
| IZO | 28.31 | -16.50 | UTA | 39.90 | -113.72 |
| KEY | 25.67 | -80.16 | UUM | 44.45 | 111.10 |
| KUM | 19.74 | -155.01 | WIS | 29.97 | 35.06 |
| KZD | 44.08 | 76.87 | WLG | 36.29 | 100.90 |
| KZM | 43.25 | 77.88 | WPC | Shipborne data | Shipborne data |
| LJO | 32.87 | -117.26 | ZEP | 78.91 | 11.89 |

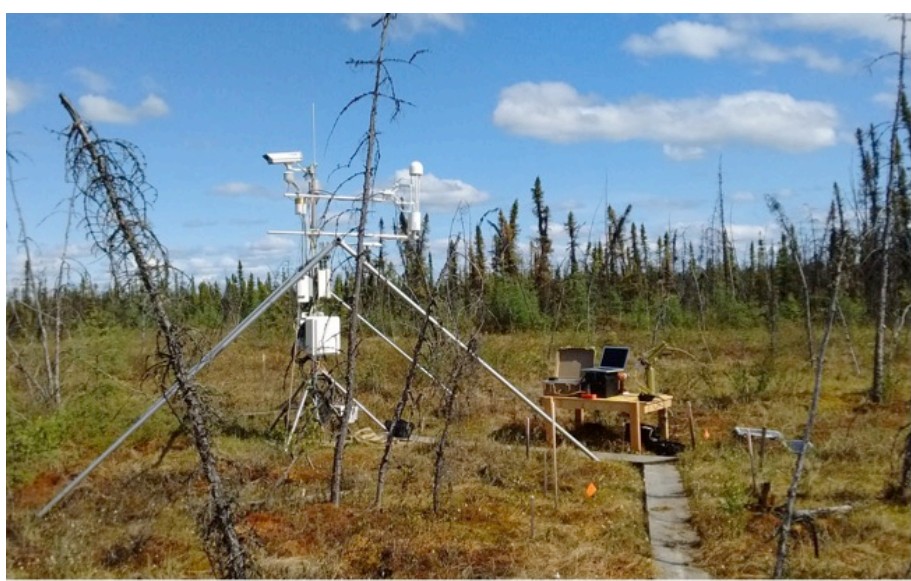

**Figure 1:** The mini-LHR (right) is portable and can be deployed to remote locations where larger TCCON installations are not possible due to the fragile ground conditions. Here, the mini-LHR monitors XCO₂ and XCH₄ alongside an eddy covariance tower (left) in a collapse scar bog permafrost site in Alaska

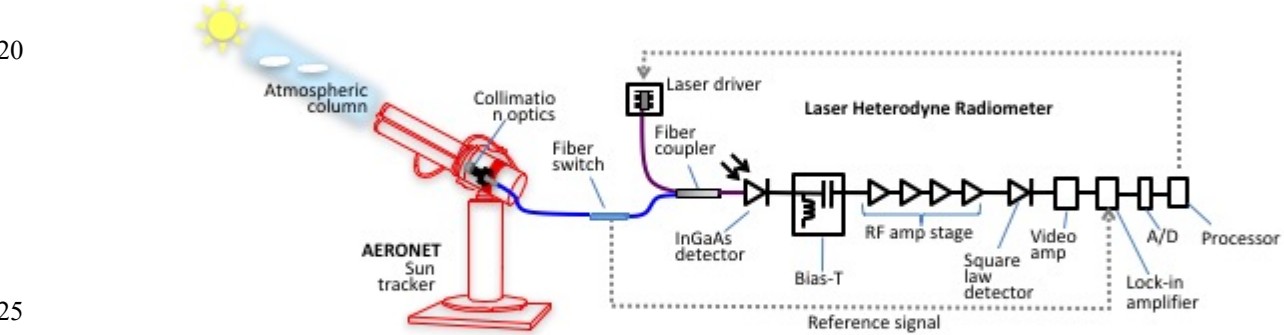

**Figure 2:** Schematic of a mini-LHR. Sunlight is collected with collection optics that are non-invasively connected to the AERONET sun tracker. Sunlight is then modulated with a fibre switch, superimposed with infrared laser light from a distributive feedback laser in a single mode fibre coupler, and mixed in a fast photoreceiver/InGaAs detector to produce a radio frequency (RF) beat signal. In the RF receiver (custom), a bias tee separates RF and DC outputs. The RF signal passes through a gain stage, and is then detected with a square-law detector. The signal is measured with a lock-in amplifier that is referenced to the modulation frequency as the laser scans across an absorption feature. A microprocessor controls the laser scanning and data collection.

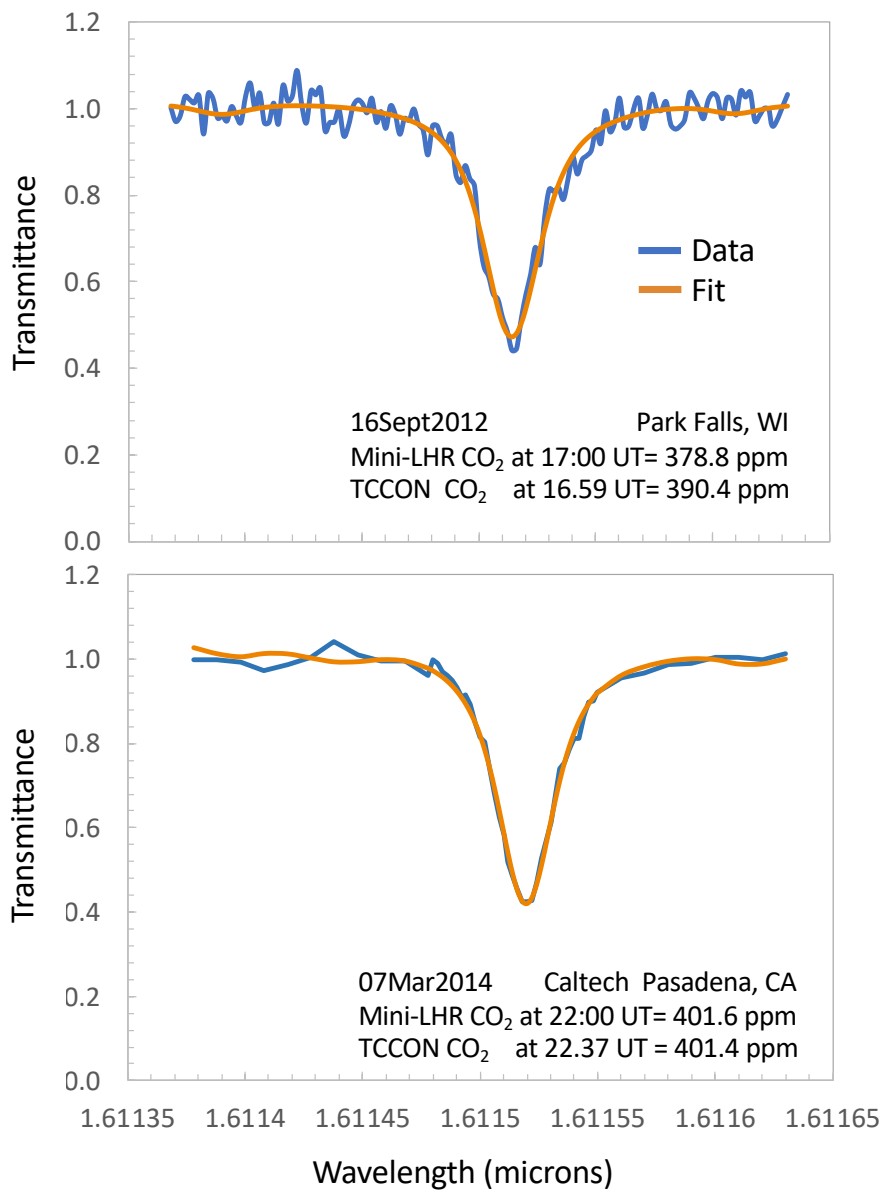

**Figure 3:** Side-by-side comparisons of mini-LHR and TCCON instruments at (top panel) Park Falls on 16 September 2012 and at (bottom panel) Caltech on 7 March 2014. Mini-LHR data is shown in blue with the PSG retrieval fit in orange. The 2012 data is the average of three scans collected over the period of an hour and the 2014 data is the average of five scans collected over the period of a half hour. The resulting XCO2 value for the mini-LHR and the nearest corresponding value for TCCON are shown inset.

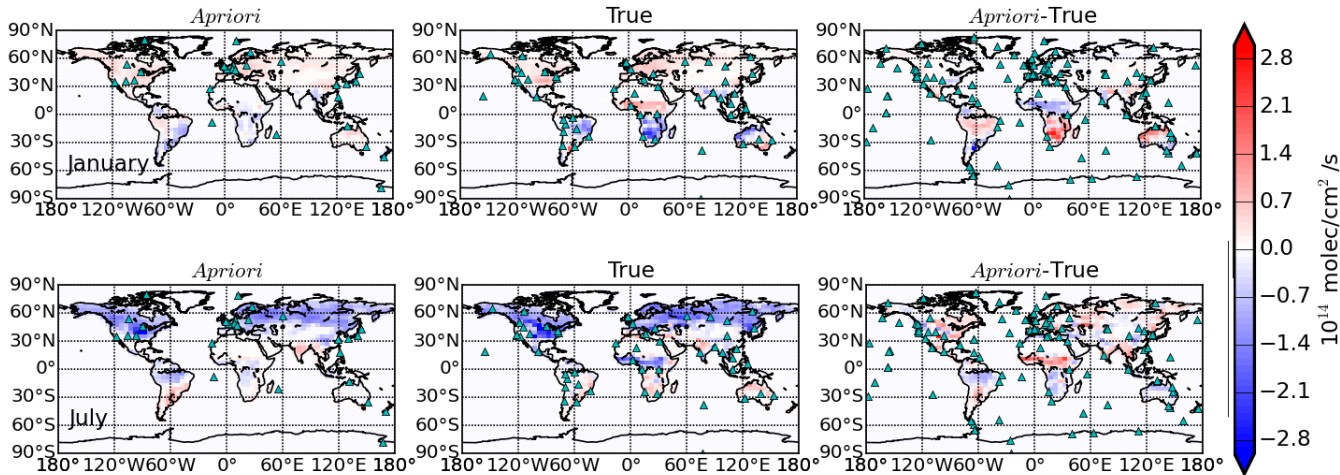

**Figure 4:** Distribution of land biosphere $CO_2$ fluxes ($10^{14}$ molec/cm$^2$/s) for January (top panels) and July (bottom panels) in our study year, described on the 4° latitude x 5° longitude GEOS-Chem model grid. The left panels show the CASA model that we used as our *a priori*; the middle panels show output from the ORCHIDEE land surface model that we use to define the true state; and the right panels show the difference between ORCHIDEE and the CASA model. The cyan triangles represent observation sites from: 1) current and future TCCON network (left panels); 2) the enhanced mini-LHR network (middle panels); 3) the subset of the NOAA ground-based network (left panels).

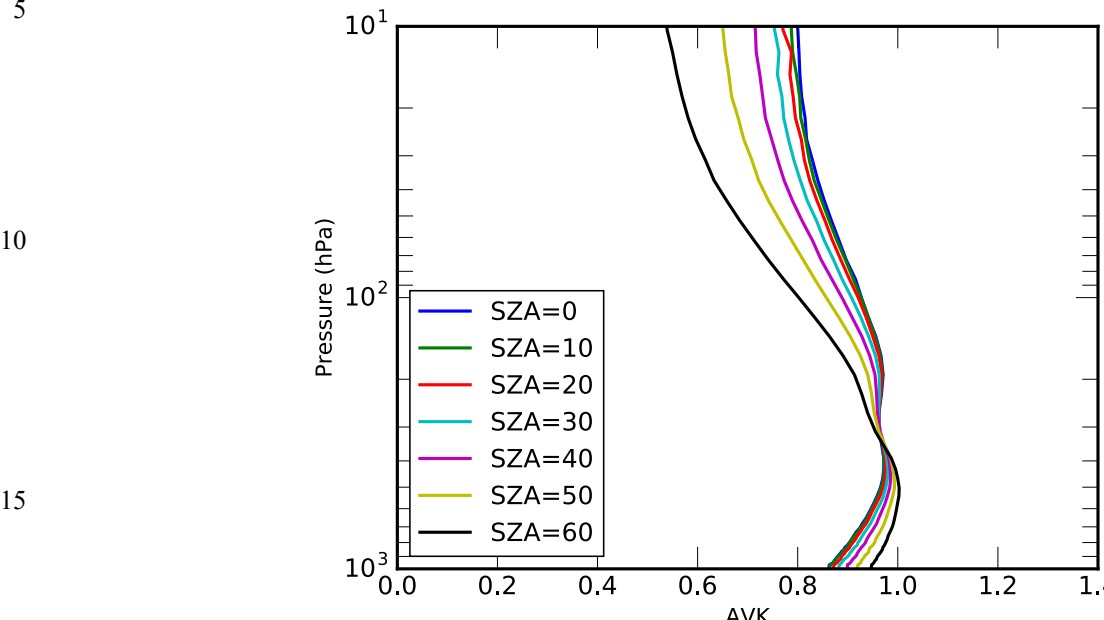

**Figure 5:** Averaging kernel ($\widehat{AK}$) values for $CO_2$ for different values of solar zenith angles. Calculations for $CO_2$ have been done assuming a mini-LHR SNR of 500 (although in reality SNR varies with SZA and is highest near a SZA of 0), and assuming a background variability of 5%.

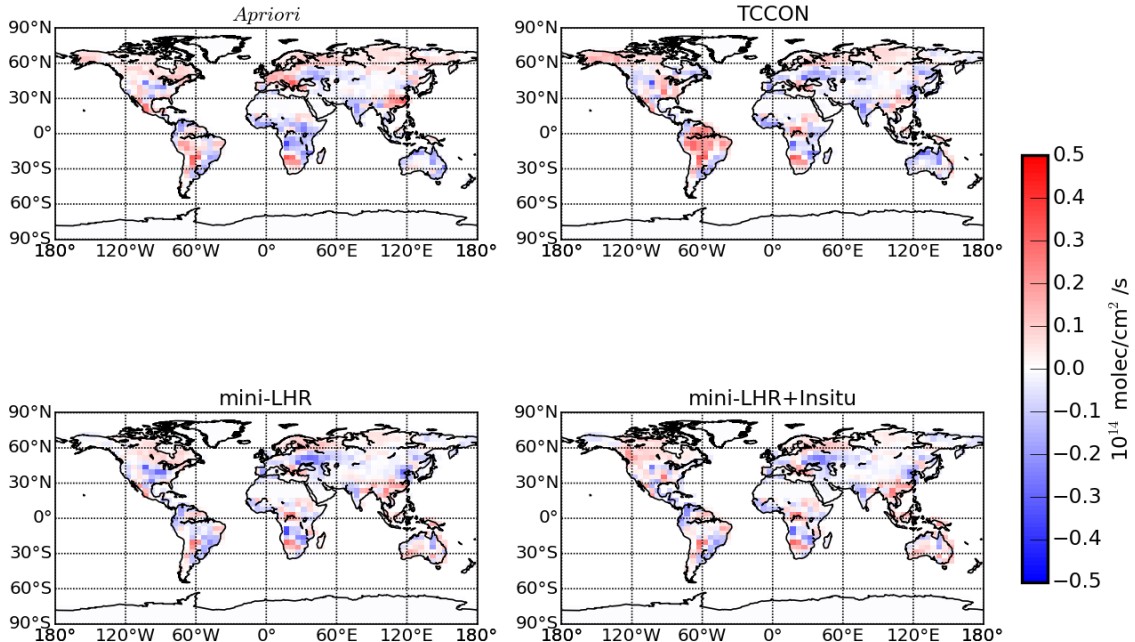

**Figure 6:** Distribution of annual mean residuals that describe the true state minus *a priori* and *a posteriori* $CO_2$ fluxes ($10^{14}$ molec/cm$^2$/s), described on the on the 4° latitude x 5° longitude GEOS-Chem model grid.

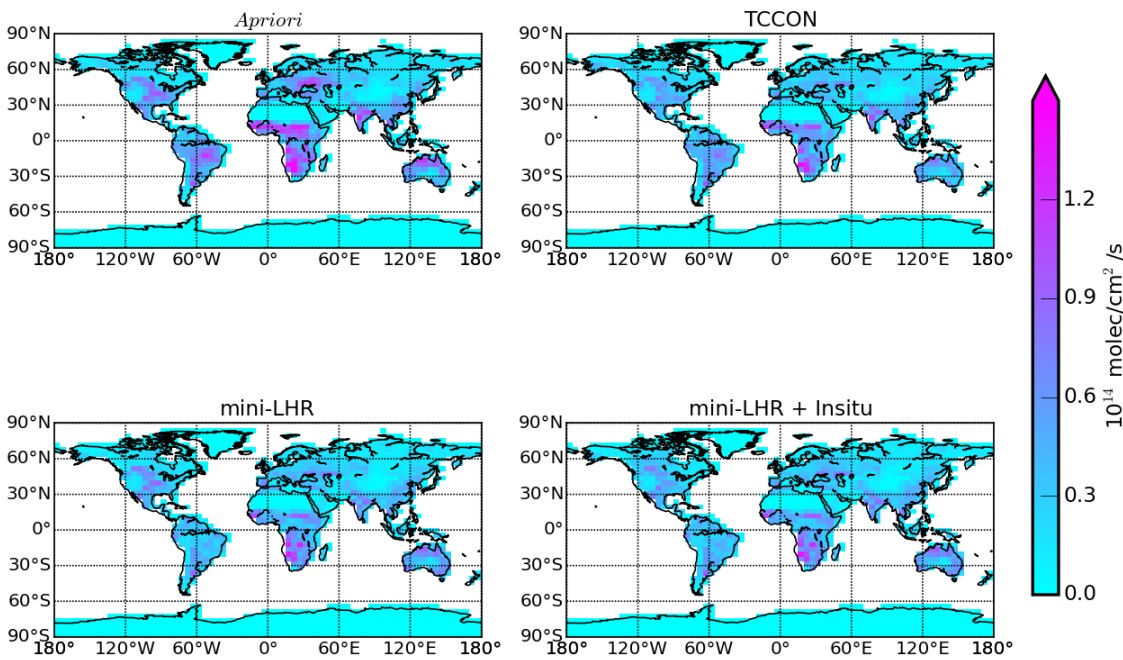

**Figure 7:** Distribution of annual mean root-mean-square-errors between the true state and *a priori* and *a posteriori* $CO_2$ fluxes ($10^{14}$ molec/cm$^2$/s), described on the on the 4$^o$ latitude x 5 $^o$ longitude GEOS-Chem model grid.

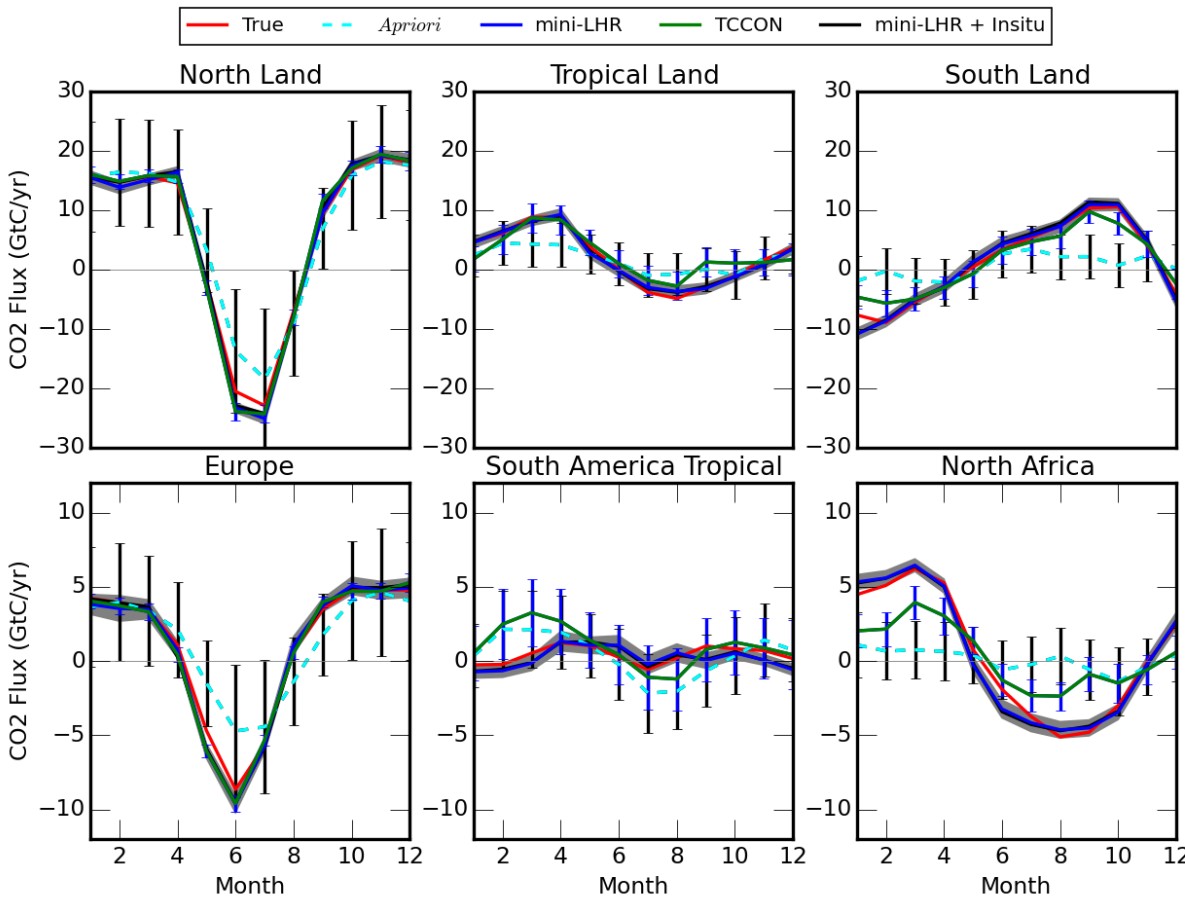

**Figure 8:** True, *a priori* and *a posteriori* monthly $CO_2$ fluxes (GtC/yr), described over large-scale geographical regions for our study year. Vertical lines and the grey envelope both denote uncertainties associated with the *a priori* fluxes and the *a posteriori* fluxes inferred from the mini-LHR+NOAA *in situ* flask networks, respectively.

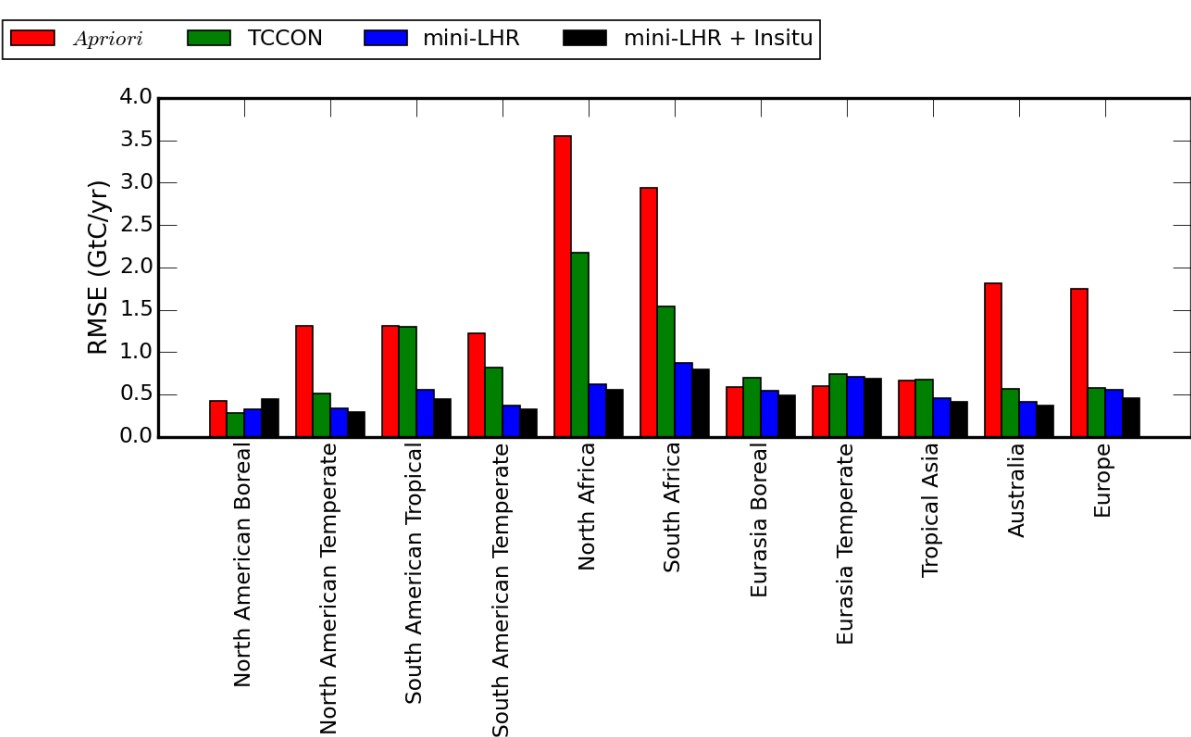

**Figure 9:** Annual mean root-mean-square errors (RMSE, GtC/yr) associated with (red) *a priori* and a *posteriori* flux estimates inferred from (green) TCCON, (blue) mini-LHR $CO_2$ columns, and from the (black) combined mini-LHR and NOAA *in situ* flask networks.

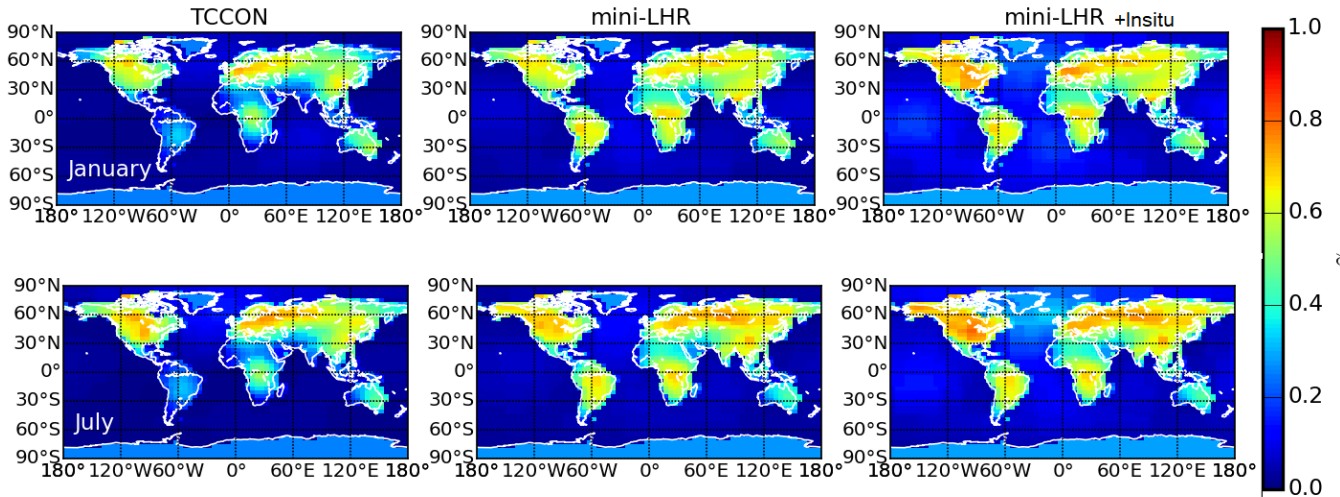

**Figure 10:** Theoretical improvement in the knowledge of $CO_2$ fluxes for a nominal (top) January and (bottom) July as determined by the $\gamma$ factor, defined in the main text, for the (left) mini-LHR, (middle) TCCON, and (right) combined mini-LHR and NOAA *in situ* flask measurement networks.