# Peer review of "Potential improvements in global carbon flux estimates from a network of laser heterodyne radiometer measurements of column carbon dioxide"

_Atmospheric Measurement Techniques, 2018_

## Referee Comment (RC1) · Anonymous Referee #1 · 27 Apr 2018

This paper describes a potential low-cost network that leverages the AERONET infrastructure (and solar tracker) to measure total column dry-air mole fractions of CO2 and CH4. The authors explore the potential for "mini-LHR" spectrometers to provide an improved understanding of CO2 fluxes using three sets of OSSEs: one using the mini-LHR network alone, one using the TCCON network alone, and one with the combined network. The results show that the mini-LHR network could significantly increase CO2 flux a posteriori knowledge.

The idea of a low-cost network to help improve the density of high-quality ground-based measurements of XCO2 is certainly an attractive one. Leveraging off of the

existing AERONET infrastructure is a great idea. The paper is generally well-written and is a suitable topic for AMT. I have several comments and requests listed below that, if addressed, would make this paper suitable for publication.

General Comments:

My main concern about this paper is the lack of consideration of site-to-site bias. This is a crucial problem in carbon cycle science, because spurious gradients in the measurements can cause us to infer large and spurious fluxes. The authors do mention a calibration of sorts using a 36-m long gas cell, but this does not seem representative of the atmosphere, changes in pressure, temperature, water vapor, and their vertical structures. Nor is there a discussion of instrument line shape (or instrument function) for these spectrometers and how much they might vary from instrument to instrument, what the airmass or solar zenith angle dependencies are likely to be, etc.

There is some discussion about ongoing side-by-side work with the Armstrong TCCON station, but there are no plots or concrete results from this work. I request that both a time series and a one-to-one plot (and any other relevant diagnostics) of the coincident measurements between the Armstrong TCCON station and the mini-LHR be presented in this paper. I also request plots of spectra and spectral fits from the coincident measurements to get a sense of the signal-to-noise ratio of the spectra and the quality of the retrievals, a priori profiles, and spectroscopy of the two retrieval algorithms.

Specific Comments:

Please provide more details of the retrieval algorithm: is it Optimal Estimation? Does the algorithm retrieve profiles of $CO_2$ or does it perform a scaling retrieval (like TCCON)? Given equation (1), I would assume the former, but it's not clear. Does the mini-LHR measure oxygen to compute the dry-air mole fractions or does it rely on a precise surface pressure and water column measurement? Plots of example spectra would be helpful.

[Figure]

P2L9: Wunch et al. 2017 wasn't pointing out how poor the data are from OCO-2, it was pointing out how good it is when you account for some known (characterizable) biases!

Please add a table of the TCCON stations used for the OSSEs. There are missing and "mystery" TCCON stations on the map in Figure 4, bottom left panel. For example, missing sites include Eureka, East Trout Lake, Hefei (which has not yet delivered data to the TCCON archive, but has published a preliminary paper: Wang, W., Tian, Y., Liu, C., Sun, Y., Liu, W., Xie, P., Liu, J., Xu, J., Morino, I., Velazco, V. A., Griffith, D. W. T., Notholt, J., and Warneke, T.: Investigating the performance of a greenhouse gas observatory in Hefei, China, Atmos. Meas. Tech., 10, 2627-2643, https://doi.org/10.5194/amt-10-2627-2017, 2017). These stations may contribute to an increase in gamma over the northern latitudes.

There are markers north of Manaus (Paramaribo?), north of Reunion Island, central Australia, and Russia (Yekaterinburg?) that are not TCCON stations. You can guide your OSSEs by the map on the TCCON wiki (https://tccon-wiki.caltech.edu/) or on the TCCON archive (https://tccondata.org/). Including the correct TCCON station locations might impact your results.

As I understand it, the gamma parameter shows the improvement of integrating the measurements into the model over the pure model uncertainties. It is interesting to note that while both OSSEs (TCCON and mini-LHR) have similar numbers of stations in Australia/New Zealand, there is little to no improvement in the uncertainties in that region. Are we to interpret from this that the models perform extremely well in that region compared with the rest of the southern hemisphere land? Please expand on this. Why are the models so good there and not elsewhere over the SH land? Should we be putting any stations in Australia/NZ at all?

How many TCCON stations would need to be added to approach the gains from 50 mini-LHR stations, and where should those TCCON stations be placed? Should they be placed where you've placed the mini-LHRs? Would fewer TCCON stations do,

if they were more strategically placed? For the same cost of the 50 mini-LHR stations, how many (if any) TCCON stations could be purchased (a TCCON station costs roughly $500,000 USD)?

In Figure 6, you show that the RMSE is more than double for the TCCON inversion than the LHR inversion over Europe? Why? There are about equal numbers of TCCON stations and LHR stations in Europe. What is it about the mini-LHR measurements in the region that provide this additional information?

In your OSSEs, what do you assume about the distribution of clouds and how they impact the density of measurements?

Technical Remarks

A 1 ppm precision after averaging over an hour ($\sim$30 measurements) is not particularly high precision these days, so please rephrase P1L16.

P1L27: Please revise the number 23 when you update your OSSEs to match the existing TCCON stations.

P2L7: GOSAT and OCO-2 measure sunlight reflected off the Earth's surface in the near-infrared, but they measure in nadir mode, glint mode, and target mode (not just nadir).

P2L11: I believe you mean to cite Wunch et al. 2011 and not 2017:

Wunch, D., G. C. Toon, J.-F. L. Blavier, R. A. Washenfelder, J. Notholt, B. J. Connor, D. W. T. Griffith, V. Sherlock, and P. O. Wennberg (2011), The Total Carbon Column Observing Network, Philos. Trans. R. Soc. A Math. Phys. Eng. Sci., 369(1943), 2087–2112, doi:10.1098/rsta.2010.0240.

P2L20: It is stated that the TCCON instruments report a precision of $\sim$1 ppm that is mitigated by comparing with aircraft profiles. This should be an *accuracy*, not a precision. The precision of TCCON is $\sim$0.4 ppm (1-sigma), according to Wunch et al.

[Figure]

2010.

P2L22: We technically cannot "calibrate" when measuring the atmosphere (as it cannot be controlled), so the phrase we use for this is to "tie" the TCCON measurements to the WMO scale.

P2L31: The TCCON can also measure in breaks between clouds and measure with a similar frequency as the mini-LHR.

P3L1: The higher latitudes are measured reasonably well by the satellites during summer, but it is correct to say that they are not well covered in winter.

P4L9: Why is the scattering package required for direct sun-viewing measurements?

P4L28: "PSG/API to *calculate* spectra" (remove "get").

P5L19: Add "a": "known to be *a* significant source".

P5L26-27: Direct sun-viewing measurements should have very high signal-to-noise ratio (because the signal is so large). Please clarify.

P7L23: Why did you adopt a "uniform 50% a priori uncertainty and 1.5 ppm for individual measurement and model transport errors"?

P7L34: "using mini-LHR measurements collected *and* enhanced measurement configuration"

P8L3: Not including instrument biases is problematic for the carbon cycle (see General Comments).

P5L28: Wunch et al. 2011 is not the correct reference for the DC correction; please cite:

Keppel-Aleks, G., G. C. Toon, P. O. Wennberg, and N. M. Deutscher (2007), Reducing the impact of source brightness fluctuations on spectra obtained by Fourier-transform spectrometry., Appl. Opt., 46(21), 4774–4779, doi:10.1364/AO.46.004774.

Figure 3: Plotting column averaging kernels on a linear pressure grid is helpful for the total column, which is weighted by mass.

Table 2: Please organize this table by the Figure 6 sectors.

Figure 6: Could you please add an extra bar for the mini-LHR+TCCON inversions?

---

## Referee Comment (RC2) · Anonymous Referee #2 · 8 Jun 2018

The authors have performed Observing System Simulation Experiments (OSSEs) to evaluate the potential of a new network of mini-LHR instruments to reduce uncertainty in carbon flux estimates. The mini-LHR is a low-cost instrument that can be installed at AERONET sites globally, potentially leading to a significant expansion of the surface observing network. The OSSEs in the manuscript showed that with only 50 mini-LHR sites, well located around the globe, it would be possible to greatly reduce $CO_2$ flux uncertainties. The paper is well written, and the suggestion of deploying the mini-LHR in tandem with the sun photometers at the AERONET sites is an excellent idea. However, I cannot recommend the paper for publication in its present form. I believe that additional OSSE work, as described below, is needed before the paper would be

[Figure]

acceptable for publication.

Main Comments

1. My first main concern is that the OSSEs do not account for spatially and temporally varying systematic errors in the synthetic data. This would have been an acceptable OSSE study a decade ago, but experience with GOSAT and OCO-2 data has shown that systematic errors are the main challenge with using XCO2 data for flux inversions. Indeed, this was noted by the authors on Page 2, where they stated that for GOSAT and OCO-2 "poorly characterized systematic errors compromise the accuracy of their data (Wunch et al., 2017) and limit the utility of such datasets for inferring surface flux distributions (Basu et al., 2013)." In light of this, I don't see how the authors can neglect systematics errors in their OSSEs. I am sure that the authors are aware of the numerous published OSSEs that were conducted before the launch of GOSAT and OCO that argued that future satellite observations of CO2 will significantly reduce flux uncertainties. Unfortunately, many of those OSSEs did not realistically look at the impact of systematic errors on the flux inversions. The authors must address this in their OSSEs before this study can be considered acceptable for publication.

2. My second main concern is with the use of TCCON as a benchmark for the OSSEs. It has been shown that TCCON can provide useful information on the carbon cycle, but the network was designed mainly for satellite validation. If the focus of this manuscript is on the "potential improvements in global carbon flux estimates" associated with the mini-LHR network, the issue should be examined in the context of the added value of the mini-LHR network given the exiting in situ and satellite observing systems that provide observations used in flux inversions. TCCON data are rarely used for flux inversions. At a minimum, the authors should have included the in situ surface network (the flask and quasi-continuous sites) in their OSSEs. However, I would prefer to see a comparison involving the in situ network and OCO-2 with the mini-LHR network.

3. Another concern that I have with the OSSEs here is that the same model is used

for the nature run as for the assimilation. A challenge with CO2 flux inversions is that we don't unknown how model errors will be manifested in the estimated fluxes given the information content of the data. Using the same model to produce the synthetic data and for the assimilation creates an overly optimistic scenario. I would encourage the authors to use output from another model, using different meteorological fields, to produce their synthetic data. This will provide a more rigorous OSSE and is now standard OSSE practice (See Hoffman and Atlas, Future Observing System Simulation Experiments, BAMS, Vol 9, 1601-1616, 2016).

4. The authors acknowledge that cloud cover is an issue, i.e. data are collected "throughout the day during sunlight hours when clouds are not present." How was that accounted for in the OSSEs? Did they use the MERRA cloud fields to simulate data loss due to cloud cover? It is unclear if this was done. Capturing this well is important for contrasting the regional improvements in the flux estimates as data loss will be worse in some regions than others, and will vary seasonally.

Minor Comments

1. Page 2, line 13: define mini-LHR.

2. Page 3, line 9: it should be "result in a" and "will be the".

3. Page 4, lines 19 and 20: Is it MERRA or MERRA-2?

4. Page 4, line 28: Some words are missing here: "to get calculate spectra".

5. Page 6, line 24: No, this is not a rigorous test of the data. Please see main Comments 1, 3, and 4 above.

6. Page 7, line 18: How reasonable is this 5% error? On what it is based? The assumed a priori error will influence the estimated DOF. An overestimate of the a priori error will result in artificially large DOFs.

7. Page 8, line 3: Instrument biases really must be included in the OSSEs (see main

comment).

8. Page 9, lines 2-4: What is the reference for this statement? It is not clear to me from what was presented that the performance shown here rivals the in situ network over North America and outperforms it in the tropics. This is why I would like to see the in situ data included in the OSSEs (see main Comment 2).

————————————————————

---

## Short Comment (SC1) · 13 Feb 2019

The model section of this study provides useful information concerning how our capabilities of estimating $CO_2$ fluxes could be improved by further extending our observational capabilities. The problem I have with this paper is due to the fact that it does not perform this examination on the grounds of a hypothetical network of selectable quality, but explicitly refers to a network comprised of a certain type of existing device (mini-LHR instruments). If such a reference is chosen, it seems essential to me to include a reliable characterization of the actual performance achieved by this kind of device. The noise error of a single measurement recorded with a single unit of a network is not at

all the critical issue when investigating what gain could be induced by adding such a network. The impact of systematic errors (between units: site-to-site biases, and correlated errors for a selected unit: drifts) is in my opinion not adequately addressed in the paper. A proper characterization of the error budget seems an essential prerequisite to this study as the authors explicitly refer to a certain kind of existing instrumentation (similar requirements would result from the claim that a mini-LHR network would be useful for satellite validation).

Therefore, the announced instrumental study should be published before the submitted paper in order to provide a reference. A longterm side-by-side comparison of one unit with a TCCON spectrometer (spanning at least one annual cycle) would be a minimum requirement. (A previous version of the manuscript handled under AMTD manuscript number 2017-368 provided slightly more information and presented actual mini-LHR spectra, so this current version of the manuscript seems to me a further degradation of the previous presentation.)

Moreover, the impression that no adequate investigation at all is attempted by the authors for establishing a reliable error budget is further consolidated by the fact that even a simple estimation of the total error budget based on plausible assumptions concerning instrument performance and a-priori knowledge of the atmospheric state is not provided. Note that TCCON uses the co-observed column of molecular oxygen for generating column-averaged dry-air mole fractions. This step is useful not only for reducing the error propagation of instrumental imperfections, but also to reduce other detrimental impacts, as e.g. errors in the assumed atmospheric temperature and ground pressure - I would therefore expect a less favourable error budget for the mini-LHR instrument.

In summary, I would require to profoundly strengthen the part on instrument characterization and overall error budget of the proposed network (or publish these aspects before the presented kind of study), and to apply the resulting correlated measurement errors (drifts, airmass-dependent effects, site-to-site biases, etc.) for achieving a real-

istic estimate of the potential improvements. I do not recommend a publication of this work in its current shape.

---

## Author Comment (AC1) · 20 Mar 2019

Reviewer #1

General Comment: My main concern about this paper is the lack of consideration of site-to-site bias. This is a crucial problem in carbon cycle science, because spurious gradients in the measurements can cause us to infer large and spurious fluxes. The authors do mention a calibration of sorts using a 36-m long gas cell, but this does not seem representative of the atmosphere, changes in pressure, temperature, water vapor, and their vertical structures. Nor is there a discussion of instrument line shape (or instrument function) for these spectrometers and how much they might vary from

instrument to instrument, what the airmass or solar zenith angle dependencies are likely to be, etc. There is some discussion about ongoing side-by-side work with the Armstrong TCCON station, but there are no plots or concrete results from this work. I request that both a time series and a one-to-one plot (and any other relevant diagnostics) of the coincident measurements between the Armstrong TCCON station and the mini-LHR be presented in this paper. I also request plots of spectra and spectral fits from the coincident measurements to get a sense of the signal-to-noise ratio of the spectra and the quality of the retrievals, a priori profiles, and spectroscopy of the two retrieval algorithms.

Response: Instrument bias will ultimately be addressed through regular side-by-side comparisons with the TCCON instrument at NASA Armstrong. There would be a "standard" mini-LHR instrument that is regularly co-located at this site that would then be compared to all other mini-LHR instruments. The purpose of the 36-m gas cell with a known NIST standard atmosphere sample is for traceability and tracking long-term instrument performance but is not intended to address bias between sites. We have completed two short-duration comparisons at TCCON sites in Park Falls, WI (2012) and at Caltech (2014) and have added sample scans and spectral fits as requested. A long-term comparison will be the focus of an instrument paper and is out of the scope of this paper which is demonstrates the potential benefit of a network, assuming of course that the instrument meets all the performance requirements.

Specific Comments: Please provide more details of the retrieval algorithm: is it Optimal Estimation? Does the algorithm retrieve profiles of $CO_2$ or does it perform a scaling retrieval (like TCCON)? Given equation (1), I would assume the former, but it's not clear. Does the mini-LHR measure oxygen to compute the dry-air mole fractions or does it rely on a precise surface pressure and water column measurement? Plots of example spectra would be helpful.

Response: We use an optimal estimation retrieval approach. The retrieval employs NASA MERRA2 meteorological analyses to define the state and a-priori values for the

atmosphere. We "perturb" the CO2 profile by a scaler, which is the value that it is actually being retrieved by the retrieval algorithm. Our retrieval uses assimilated meteorological fields of pressure, temperature, water vapor, ozone, and water ice clouds from the surface to ∼80 km (72 layers) with a cadence of 180 minutes, and spatial resolution of ∼0.5 degrees (576 x 361). The values are further refined temporally and spatially to a resolution of better than 1 km employing the USGS-GTOPO30 topographic maps and considering a hydrostatic equilibrated atmosphere within every bin. We have added text and a reference to the retrieval approach. We have also provided figures showing sample spectra using the retrieval fit.

P2L9: Wunch et al. 2017 wasn't pointing out how poor the data are from OCO-2, it was pointing out how good it is when you account for some known (characterizable) biases!

Response: Poorly characterized systematic errors are progressively less of an issue in the interpretation of GOSAT and OCO-2 XCO2 data but they do remain. Wunch et al, 2017 state "After bias correction, residual biases remain. These biases appear to depend on latitude, surface properties, and scattering by aerosols." These residual biases are considered to be uncharacterized. "Remedying these residual biases is the current focus of the OCO-2 algorithm development and validation teams, and we anticipate that the next version of the OCO-2 data will represent a significant improvement." We have toned down this text.

Please add a table of the TCCON stations used for the OSSEs. There are missing and "mystery" TCCON stations on the map in Figure 4, bottom left panel. For example, missing sites include Eureka, East Trout Lake, Hefei (which has not yet delivered data to the TCCON archive, but has published a preliminary paper: Wang, W., Tian, Y., Liu, C., Sun, Y., Liu, W., Xie, P., Liu, J., Xu, J., Morino, I., Velazco, V. A., Griffith, D. W. T., Notholt, J., and Warneke, T.: Investigating the performance of a greenhouse gas observatory in Hefei, China, Atmos. Meas. Tech., 10, 2627-2643, https://doi.org/10.5194/amt-10-2627-2017, 2017). These stations may contribute to an increase in gamma over the northern latitudes.

Response: Thank you for pointing out this oversight, we have added a table of TCCON sites used in this study.

There are markers north of Manaus (Paramaribo?), north of Reunion Island, central Australia, and Russia (Yekaterinburg?) that are not TCCON stations. You can guide your OSSEs by the map on the TCCON wiki (https://tccon-wiki.caltech.edu/) or on the TCCON archive (https://tccondata.org/). Including the correct TCCON station locations might impact your results.

Response: We find that additional sites generally improve observation constraints by the TCCON network, but they have not significantly changed our conclusions.

As I understand it, the gamma parameter shows the improvement of integrating the measurements into the model over the pure model uncertainties. It is interesting to note that while both OSSEs (TCCON and mini-LHR) have similar numbers of stations in Australia/New Zealand, there is little to no improvement in the uncertainties in that region. Are we to interpret from this that the models perform extremely well in that region compared with the rest of the southern hemisphere land? Please expand on this. Why are the models so good there and not elsewhere over the SH land? Should we be putting any stations in Australia/NZ at all?

Response: The prior uncertainty is assumed to be 50% of NEE. As a result, over a large part of Australia and New Zealand, (assumed) prior flux uncertainties are small compared to northern mid-latitude regions. This is further complicated by our limited model resolution (4ox5o).

How many TCCON stations would need to be added to approach the gains from 50 mini-LHR stations, and where should those TCCON stations be placed? Should they be placed where you've placed themini-LHRs? Would fewer TCCON stations do,if they were more strategically placed? For the same cost of the 50 mini-LHR stations, how many (if any) TCCON stations could be purchased (a TCCON station costs roughly $500,000 USD)?

Response: These are all good questions, but they lie outside the scope of this paper. The focus of this paper was to exploit the complementarity of the instruments, accounting for their (dis)advantages. An economic argument is not helpful given the sparseness of the existing atmospheric $CO_2$ and $CH_4$ network. Given that TCCON sites are 50 times more expensive than the mini-LHR it seems that TCCON would not be able to compete, especially given the portability of the smaller instruments. BUT TCCON sites remain invaluable given their accuracy and precision performance.

In Figure 6, you show that the RMSE is more than double for the TCCON inversion than the LHR inversion over Europe? Why? There are about equal numbers of TCCON stations and LHR stations in Europe. What is it about the mini-LHR measurements in the region that provide this additional information?

Response:Over some regions the LHR network performances better, reflecting their geographical locations that are more sensitive to regional outflow where there are large differences between the CASA and ORCHIDEE models.

In your OSSEs, what do you assume about the distribution of clouds and how they impact the density of measurements?

Response:We determine clear sky measurements the same way for TCCON and LHR: by randomly sampling cloud coverage from ECMWF-interim reanalysis. The resulting measurement density reflects the probability of cloud-free scenes at model grids. We acknowledge there are many unaccounted issues in this approach, e.g. the influence of cloud 3D distributions.

Technical Remarks

A 1 ppm precision after averaging over an hour (30 measurements) is not particularly high precision these days, so please rephrase P1L16.

Response:This has been rephrased.

P1L27: Please revise the number 23 when you update your OSSEs to match the

existing TCCON stations.

Response:Thanks for the suggestion, we have now recomputed the OSSEs using the revised TCCON network configuration.

P2L7: GOSAT and OCO-2 measure sunlight reflected off the Earth's surface in the near-infrared, but they measure in nadir mode, glint mode, and target mode (not just nadir).

Response:Agreed, nadir mode is the closest comparison to the TCCON and mini-LHR observations. We will add this detail.

P2L11: I believe you mean to cite Wunch et al. 2011 and not 2017: Wunch, D., G. C. Toon, J.-F. L. Blavier, R. A. Washenfelder, J. Notholt, B. J. Connor, D. W. T. Griffith, V. Sherlock, and P. O. Wennberg (2011), The Total Carbon Column Observing Network, Philos. Trans. R. Soc. A Math. Phys. Eng. Sci., 369(1943), 2087–2112, doi:10.1098/rsta.2010.0240.

Response:Thanks for catching this typo.

P2L20: It is stated that the TCCON instruments report a precision of 1 ppm that is mitigated by comparing with aircraft profiles. This should be an *accuracy*, not a precision. The precision of TCCON is 0.4 ppm (1-sigma), according to Wunch et al. 2010.

Response:This has been corrected in the text.

P2L22: We technically cannot "calibrate" when measuring the atmosphere (as it cannot be controlled), so the phrase we use for this is to "tie" the TCCON measurements to the WMO scale.

Response:Understood. This has been corrected in the text.

P2L31: The TCCON can also measure in breaks between clouds and measure with a similar frequency as the mini-LHR.

Response:This has been clarified in the text.

P3L1: The higher latitudes are measured reasonably well by the satellites during summer, but it is correct to say that they are not well covered in winter.

Response:Agreed. We have clarified that point in the text.

P4L9: Why is the scattering package required for direct sun-viewing measurements?

Response:Our scattering package is used because it also includes the treatment of aerosols, which we use to properly model the continuum shape. We only use the "extinction" component of this package (not the scattering part, N-stream pairs = 0).

P4L28: "PSG/API to *calculate* spectra" (remove "get").

Response:Corrected

P5L19: Add "a": "known to be *a* significant source".

Response:Corrected

P5L26-27: Direct sun-viewing measurements should have very high signal-to-noise ratio (because the signal is so large). Please clarify.

Response:This text has been reworded.

P7L23: Why did you adopt a "uniform 50% a priori uncertainty and 1.5 ppm for individual measurement and model transport errors"?

Response:Uncertainties of biosphere models are still not fully quantified. For simplicity we have assumed n uniform 50% a priori uncertainty, following many previous studies. Similarly, we assume a uniform model error. Quantification of observation uncertainty, and particularly systematic errors, is still on-going. Here we assume a conservative value for observation errors based on our field experiments.

P7L34: "using mini-LHR measurements collected *and* enhanced measurement configuration"

Response:Corrected

P8L3: Not including instrument biases is problematic for the carbon cycle (see General Comments).

Response:This has been addressed earlier in responses to general comments.

P5L28: Wunch et al. 2011 is not the correct reference for the DC correction; please cite: Keppel-Aleks, G., G. C. Toon, P. O. Wennberg, and N. M. Deutscher (2007), Reducing the impact of source brightness fluctuations on spectra obtained by Fourier-transform spectrometry., Appl. Opt., 46(21), 4774–4779, doi:10.1364/AO.46.004774.

Response:Thank you, this has been corrected.

Figure 3: Plotting column averaging kernels on a linear pressure grid is helpful for the total column, which is weighted by mass.

Response:Plot has been changed

Table 2: Please organize this table by the Figure 6 sectors.

Response:Table 2 has been changed, as suggested

Figure 6: Could you please add an extra bar for the mini-LHR+TCCON inversions?

Response:Figure 6 has been changed, as we now show the mini-LHR+NOAA insitu sites, as suggested by the reviewers.

Reviewer #2

1. My first main concern is that the OSSEs do not account for spatially and temporally varying systematic errors in the synthetic data. This would have been an acceptable OSSE study a decade ago, but experience with GOSAT and OCO-2 data has shown that systematic errors are the main challenge with using XCO2 data for flux inversions. Indeed, this was noted by the authors on Page 2, where they stated that for GOSAT and OCO-2 "poorly characterized systematic errors compromise the accuracy of their

data (Wunch et al., 2017) and limit the utility of such datasets for inferring surface flux distributions (Basu et al., 2013)." In light of this, I don't see how the authors can neglect systematics errors in their OSSEs. I am sure that the authors are aware of the numerous published OSSEs that were conducted before the launch of GOSAT and OCO that argued that future satellite observations of CO2 will significantly reduce flux uncertainties. Unfortunately, many of those OSSEs did not realistically look at the impact of systematic errors on the flux inversions. The authors must address this in their OSSEs before this study can be considered acceptable for publication.

Response:We appreciate this comment, but the focus of the paper is to determine the relative importance of mini-LHR against the TCCON instruments. Both sets of synthetic data are treated the same so their ability to determine regional CO2 fluxes can be compared. Poorly characterized systematic errors are progressively less of an issue in the interpretation of GOSAT and OCO-2 XCO2 data but they do remain. The newest versions of OCO-2 data have made great strides in minimizing systematic errors. With respect to this reviewer, we are unsure what this study would gain from including a description of systematic errors and then applying a bias correction, without any justification for the size and nature of the systematic error.

2. My second main concern is with the use of TCCON as a benchmark for the OSSEs. It has been shown that TCCON can provide useful information on the carbon cycle, but the network was designed mainly for satellite validation. If the focus of this manuscript is on the "potential improvements in global carbon flux estimates" associated with the mini-LHR network, the issue should be examined in the context of the added value of the mini-LHR network given the exiting in situ and satellite observing systems that provide observations used in flux inversions. TCCON data are rarely used for flux inversions. At a minimum, the authors should have included the in situ surface network (the flask and quasi-continuous sites) in their OSSEs. However, I would prefer to see a comparison involving the in situ network and OCO-2 with the mini-LHR network.

Response:Point well taken. As the reviewer will be aware, the ground-based networks

were designed to sample large spatial scales and long temporal scales, and have an uneven distribution mainly focused on northern hemispheric maritime regions. We have now included calculations that describe the relative performance of the ground-based in situ network, as suggested. We fully agree (and acknowledge in the paper) that multiple data streams are necessary to infer surface fluxes at various temporal and spatial scales. Our proposed LHR network is designed primarily to help validate satellite measurements, but of course will be able to contribute to existing and future measurement networks.

3. Another concern that I have with the OSSEs here is that the same model is used for the nature run as for the assimilation. A challenge with CO2 flux inversions is that we don't unknown how model errors will be manifested in the estimated fluxes given the information content of the data. Using the same model to produce the synthetic data and for the assimilation creates an overly optimistic scenario. I would encourage the authors to use output from another model, using different meteorological fields, to produce their synthetic data. This will provide a more rigorous OSSE and is now standard OSSE practice (See Hoffman and Atlas, Future Observing System Simulation Experiments, BAMS, Vol 9, 1601-1616, 2016).

Response:We did use the same atmospheric transport model for the nature and assimilation runs but we used independent biospheric fluxes from the ORCHIDEE and CASA land biosphere models. ORCHIDEE and CASA CO2 fluxes are very different in seaonal magnitude and distribution, so we believe this is a good test for the simulated data. Our calculations are not focused on understanding the influence of model atmospheric transport error but are intended to assess the relative performance of a network of mini-LHR instruments and the current TCCON. We have added a caveat in the methods and conclusion about results being a conservative approach. Please note we have intentionally focused on the unitless gamma metric, associated with the relative improvement in CO2 flux uncertainty, rather than only report absolute values that could be misconstrued, considering their dependence on the assumed true flux

(or more precisely, on the assumed differences between the 'true' and the a priori).

4. The authors acknowledge that cloud cover is an issue, i.e. data are collected "throughout the day during sunlight hours when clouds are not present." How was that accounted for in the OSSEs? Did they use the MERRA cloud fields to simulate data loss due to cloud cover? It is unclear if this was done. Capturing this well is important for contrasting the regional improvements in the flux estimates as data loss will be worse in some regions than others, and will vary seasonally.

Response:Clear-sky measurements are determined by randomly sampling cloud coverage taken from ECMWF ERA-interim reanalysis. We agree that there are a lot of issues associated with such a simple approach to determine complicated cloud contamination probability, due to, for example, coarse spatial and temporal resolution of model cloud coverage relative to instrument view spots.

Minor comments

1. Page 2, line 13: define mini-LHR.

Response:This text has been added.

2. Page 3, line 9: it should be "result in a" and "will be the".

Response:This has been corrected.

3. Page 4, lines 19 and 20: Is it MERRA or MERRA-2?

Response:It is MERRA-2. The text has been corrected.

4. Page 4, line 28: Some words are missing here: "to get calculate spectra".

Response:This has been corrected.

5. Page 6, line 24: No, this is not a rigorous test of the data. Please see main Comments 1, 3, and 4 above.

Response:Please see above responses.

6. Page 7, line 18: How reasonable is this 5% error? On what it is based? The assumed a priori error will influence the estimated DOF. An overestimate of the a priori error will result in artificially large DOFs.

Response:5% of CO2 will be 20 ppm, which is larger than typical discrepancies between models and data.

7. Page 8, line 3: Instrument biases really must be included in the OSSEs (see main comment).

Response:Please see our responses above.

8. Page 9, lines 2-4: What is the reference for this statement? It is not clear to me from what was presented that the performance shown here rivals the in situ network over North America and outperforms it in the tropics. This is why I would like to see the in situ data included in the OSSEs (see main Comment 2).

Response:We have now included the relative performance of the in situ network.

Frank Hase

The model section of this study provides useful information concerning how our capabilities of estimating CO2 fluxes could be improved by further extending our observational capabilities. The problem I have with this paper is due to the fact that it does not perform this examination on the grounds of a hypothetical network of selectable quality, but explicitly refers to a network comprised of a certain type of existing device (mini-LHR instruments). If such a reference is chosen, it seems essential to me to include a reliable characterization of the actual performance achieved by this kind of device. The noise error of a single measurement recorded with a single unit of a network is not at all the critical issue when investigating what gain could be induced by adding such a network. The impact of systematic errors (between units: site-to-site biases, and correlated errors for a selected unit: drifts) is in my opinion not adequately addressed in the paper. A proper characterization of the error budget seems an essential prerequisite

to this study as the authors explicitly refer to a certain kind of existing instrumentation (similar requirements would result from the claim that a mini-LHR network would be useful for satellite validation).

Response:This point is well taken. As this reviewer will appreciate long-term drift and other sources of systematic error are generally difficult to assess within a global network. We have assumed zero systematic error for each of the studied measurement networks studied but acknowledge they do exist at some level albeit smaller for the more established, better characterized networks. Making this assumption allows us to compare the ability of complementary network to quantify regional $CO_2$ fluxes. To address this reviewer comment, we have added a paragraph that outlines this point on page 7 and on page 11.

Therefore, the announced instrumental study should be published before the submitted paper in order to provide a reference. A longterm side-by-side comparison of one unit with a TCCON spectrometer (spanning at least one annual cycle) would be a minimum requirement. (A previous version of the manuscript handled under AMTD manuscript number 2017-368 provided slightly more information and presented actual mini-LHR spectra, so this current version of the manuscript seems to me a further degradation of the previous presentation.)

Response:This version of the paper does include spectra and information about the mini-LHR, included as a response to comments in the review process. Realistically, we cannot include a long-term side-by-side comparison of the mini-LHR TCCON in a time fashion. In any case, having that additional information would not substantively improve what is essentially a model-led paper. The main message of the paper, which this reviewer will appreciate, is that ground-based remote sensing of $CO_2$ (whether that is achieved using LHR or FTIR technology) can and will play an important role in understanding the global carbon cycle: directly by providing constraints on atmospheric columns and indirectly by providing ground-truth anchor points for satellite instruments.

[Figure]

Moreover, the impression that no adequate investigation at all is attempted by the authors for establishing a reliable error budget is further consolidated by the fact that even a simple estimation of the total error budget based on plausible assumptions concerning instrument performance and a-priori knowledge of the atmospheric state is not provided. Note that TCCON uses the co-observed column of molecular oxygen for generating column-averaged dry-air mole fractions. This step is useful not only for reducing the error propagation of instrumental imperfections, but also to reduce other detrimental impacts, as e.g. errors in the assumed atmospheric temperature and ground pressure - I would therefore expect a less favourable error budget for the mini-LHR instrument.

Response:This reviewer will appreciate the modelling study reported here builds on studies that have already been published that include an instrument design paper (Wilson et al, 2013; doi: 10.1007/s00340-013-5531-1), a detailed error analysis (Clarke et al, 2014; doi:10.1088/0957-0233/25/5/055204), a description of autonomous field measurements (Melroy et al, 2015, doi: 10.1007/s00340-015-6172-3). Collectively, these studies provide some background on the technology and previous deployments of the instrument. These papers are cited in the current study.

In summary, I would require to profoundly strengthen the part on instrument characterization and overall error budget of the proposed network (or publish these aspects before the presented kind of study), and to apply the resulting correlated measurement errors (drifts, airmass-dependent effects, site-to-site biases, etc.) for achieving a realistic estimate of the potential improvements. I do not recommend a publication of this work in its current shape.

Response:Respectively, we disagree with this viewpoint. We have responded above to the individual comments.

---

## Author Response (AR2)

We thank Frank Hase and two anonymous reviewers for providing useful comments that have helped to clarify the work we present. Below we respond to individual reviewer comments.

**\section*{Reviewer #1}**

**\subsection*{General Comment:}**
*{\em My main concern about this paper is the lack of consideration of site-to-site bias. This is a crucial problem in carbon cycle science, because spurious gradients in the measurements can cause us to infer large and spurious fluxes. The authors do mention a calibration of sorts using a 36-m long gas cell, but this does not seem representative of the atmosphere, changes in pressure, temperature, water vapor, and their vertical structures. Nor is there a discussion of instrument line shape (or instrument function) for these spectrometers and how much they might vary from instrument to instrument, what the airmass or solar zenith angle dependencies are likely to be, etc. There is some discussion about ongoing side-by-side work with the Armstrong TCCON station, but there are no plots or concrete results from this work. I request that both a time series and a one-to-one plot (and any other relevant diagnostics) of the coincident measurements between the Armstrong TCCON station and the mini-LHR be presented in this paper. I also request plots of spectra and spectral fits from the coincident measurements to get a sense of the signal-to-noise ratio of the spectra and the quality of the retrievals, a priori profiles, and spectroscopy of the two retrieval algorithms.}*

Instrument bias will ultimately be addressed through regular side-by-side comparisons with the TCCON instrument at NASA Armstrong. There would be a "standard" mini-LHR instrument that is regularly co-located at this site that would then be compared to all other mini-LHR instruments.  The purpose of the 36-m gas cell with a known NIST standard atmosphere sample is for traceability and tracking long-term instrument performance but is not intended to address bias between sites. We have completed two short-duration comparisons at TCCON sites in Park Falls, WI (2012) and at Caltech (2014) and for this paper we have added sample scans and spectral fits (Figure 3 and accompanying text on page 5) as requested.  A long-term comparison will be the focus of an instrument paper and is out of the scope of this paper which is demonstrates the potential benefit of a network, assuming of course that the instrument meets all the performance requirements.

**\subsection*{Specific Comments}**
*{\em Please provide more details of the retrieval algorithm: is it Optimal Estimation? Does the algorithm retrieve profiles of CO2 or does it perform a scaling retrieval (like TCCON)? Given equation (1), I would assume the former, but it's not clear. Does the mini-LHR measure oxygen to compute the dry-air mole fractions or does it rely on a precise surface pressure and water column measurement? Plots of example spectra would be helpful.}*

We use an optimal estimation retrieval approach. The retrieval employs NASA MERRA2 meteorological analyses to define the state and a-priori values for the atmosphere. We "perturb" the $CO_2$ profile by a scaler, which is the value that it is actually being retrieved by the retrieval algorithm. Our retrieval uses assimilated meteorological fields of pressure, temperature, water

vapor, ozone, and water ice clouds from the surface to ~80 km (72 layers) with a cadence of 180 minutes, and spatial resolution of ~0.5 degrees (576 x 361). The values are further refined temporally and spatially to a resolution of better than 1 km employing the USGS-GTOPO30 topographic maps and considering a hydrostatic equilibrated atmosphere within every bin. We have added text and a reference to the retrieval approach. We have also provided figures showing sample spectra using the retrieval fit (Figure 3 and accompanying text on page 5).

*{\em P2L9: Wunch et al. 2017 wasn't pointing out how poor the data are from OCO-2, it was pointing out how good it is when you account for some known (characterizable) biases!}*

Poorly characterized systematic errors are progressively less of an issue in the interpretation of GOSAT and OCO-2 XCO2 data but they do remain. Wunch et al, 2017 state "After bias correction, residual biases remain. These biases appear to depend on latitude, surface properties, and scattering by aerosols." These residual biases are considered to be uncharacterized. "Remedying these residual biases is the current focus of the OCO-2 algorithm development and validation teams, and we anticipate that the next version of the OCO-2 data will represent a significant improvement." We have toned down this text on page 1.

*{\em Please add a table of the TCCON stations used for the OSSEs. There are missing and "mystery" TCCON stations on the map in Figure 4, bottom left panel. For example, missing sites include Eureka, East Trout Lake, Hefei (which has not yet delivered data to the TCCON archive, but has published a preliminary paper: Wang, W., Tian, Y., Liu, C., Sun, Y., Liu, W., Xie, P., Liu, J., Xu, J., Morino, I., Velazco, V. A., Griffith, D. W. T., Notholt, J., and Warneke, T.: Investigating the performance of a greenhouse gas observatory in Hefei, China, Atmos. Meas. Tech., 10, 2627-2643, https://doi.org/10.5194/amt-10-2627-2017, 2017). These stations may contribute to an increase in gamma over the northern latitudes.}*

Thank you for pointing out this oversight, we have added Table 3 that reports the TCCON sites used in this study.

*{\em There are markers north of Manaus (Paramaribo?), north of Reunion Island, central Australia, and Russia (Yekaterinburg?) that are not TCCON stations. You can guide your OSSEs by the map on the TCCON wiki (https://tccon-wiki.caltech.edu/) or on the TCCON archive (https://tccondata.org/). Including the correct TCCON station locations might impact your results.}*

We find that additional sites generally improve observation constraints by the TCCON network, but they have not significantly changed our conclusions.

*{\em As I understand it, the gamma parameter shows the improvement of integrating the measurements into the model over the pure model uncertainties. It is interesting to note that while both OSSEs (TCCON and mini-LHR) have similar numbers of stations in Australia/New Zealand, there is little to no improvement in the uncertainties in that region. Are we to interpret from this that the models perform extremely well in that region compared with the rest of the*

*southern hemisphere land? Please expand on this. Why are the models so good there and not elsewhere over the SH land? Should we be putting any stations in Australia/NZ at all?}*

The prior uncertainty is assumed to be 50% of NEE. As a result, over a large part of Australia and New Zealand, (assumed) prior flux uncertainties are small compared to northern mid-latitude regions. This is further complicated by our limited model resolution ($4^{o}$x$5^{o}$).

*{\em How many TCCON stations would need to be added to approach the gains from 50 mini-LHR stations, and where should those TCCON stations be placed? Should they be placed where you've placed themini-LHRs? Would fewer TCCON stations do,if they were more strategically placed? For the same cost of the 50 mini-LHR stations, how many (if any) TCCON stations could be purchased (a TCCON station costs roughly $500,000 USD)?}*

These are all good questions, but they lie outside the scope of this paper. The focus of this paper was to exploit the complementarity of the instruments, accounting for their (dis)advantages. An economic argument is not helpful given the sparseness of the existing atmospheric CO2 and CH4 network. Given that TCCON sites are 50 times more expensive than the mini-LHR it seems that TCCON would not be able to compete, especially given the portability of the smaller instruments. BUT TCCON sites remain invaluable given their accuracy and precision performance.

*{\em In Figure 6, you show that the RMSE is more than double for the TCCON inversion than the LHR inversion over Europe? Why? There are about equal numbers of TCCON stations and LHR stations in Europe. What is it about the mini-LHR measurements in the region that provide this additional information?}*

Over some regions the LHR network performances better, reflecting their geographical locations that are more sensitive to regional outflow where there are large differences between the CASA and ORCHIDEE models.

*{\em In your OSSEs, what do you assume about the distribution of clouds and how they impact the density of measurements?}*

We determine clear sky measurements the same way for TCCON and LHR: by randomly sampling cloud coverage from ECMWF-interim reanalysis. The resulting measurement density reflects the probability of cloud-free scenes at model grids. We acknowledge there are many unaccounted issues in this approach, e.g. the influence of cloud 3D distributions.

*\subsection*{Technical Remarks}*

*{\em A 1 ppm precision after averaging over an hour (30 measurements) is not particularly high precision these days, so please rephrase P1L16.}*

This has been edited on page 1.

*{\em P1L27: Please revise the number 23 when you update your OSSEs to match the existing TCCON stations.}*

Thanks for the suggestion, we have now recomputed the OSSEs using the revised TCCON network configuration (Table 3).

*{\em P2L7: GOSAT and OCO-2 measure sunlight reflected off the Earth's surface in the near-infrared, but they measure in nadir mode, glint mode, and target mode (not just nadir).}*

Agreed, nadir mode is the closest comparison to the TCCON and mini-LHR observations. We have now addressed this point in page 1.

*{\em P2L11: I believe you mean to cite Wunch et al. 2011 and not 2017: Wunch, D., G. C. Toon, J.-F. L. Blavier, R. A. Washenfelder, J. Notholt, B. J. Connor, D. W. T. Griffith, V. Sherlock, and P. O. Wennberg (2011), The Total Carbon Column Observing Network, Philos. Trans. R. Soc. A Math. Phys. Eng. Sci., 369(1943), 2087–2112, doi:10.1098/rsta.2010.0240.}*

Thanks for catching this typo.

*{\em P2L20: It is stated that the TCCON instruments report a precision of 1 ppm that is mitigated by comparing with aircraft profiles. This should be an \*accuracy\*, not a precision. The precision of TCCON is 0.4 ppm (1-sigma), according to Wunch et al. 2010.}*

This has been corrected in the text.

*{\em P2L22: We technically cannot "calibrate" when measuring the atmosphere (as it cannot be controlled), so the phrase we use for this is to "tie" the TCCON measurements to the WMO scale.}*

Understood. This has been corrected in the text.

*{\em P2L31: The TCCON can also measure in breaks between clouds and measure with a similar frequency as the mini-LHR.}*

This has been clarified in the text.

{\em P3L1: The higher latitudes are measured reasonably well by the satellites during summer, but it is correct to say that they are not well covered in winter.}

Agreed. We have clarified that point in the text.

{\em P4L9: Why is the scattering package required for direct sun-viewing measurements?}

Our scattering package is used because it also includes the treatment of aerosols, which we use to properly model the continuum shape. We only use the "extinction" component of this package (not the scattering part, N-stream pairs = 0).

*{\em P4L28: "PSG/API to \*calculate\* spectra" (remove "get").}*

Corrected

*{\em P5L19: Add "a": "known to be \*a\* significant source".}*

Corrected

*{\em P5L26-27: Direct sun-viewing measurements should have very high signal-to-noise ratio (because the signal is so large). Please clarify.}*

This text has been reworded.

*{\em P7L23: Why did you adopt a "uniform 50% a priori uncertainty and 1.5 ppm for individual measurement and model transport errors"?}*

Uncertainties of biosphere models are still not fully quantified. For simplicity we have assumed n uniform 50% a priori uncertainty, following many previous studies. Similarly, we assume a uniform model error.  Quantification of observation uncertainty, and particularly systematic errors, is still on-going.  Here we assume a conservative value for observation errors based on our field experiments.

*{\em P7L34: "using mini-LHR measurements collected \*and\* enhanced measurement configuration"}*

Corrected

*{\em P8L3: Not including instrument biases is problematic for the carbon cycle (see General Comments).}*

This has been addressed earlier in responses to general comments.

*{\em P5L28: Wunch et al. 2011 is not the correct reference for the DC correction; please cite:*
*Keppel-Aleks, G., G. C. Toon, P. O. Wennberg, and N. M. Deutscher (2007), Reducing the impact of source brightness fluctuations on spectra obtained by Fourier-transform spectrometry., Appl. Opt., 46(21), 4774–4779, doi:10.1364/AO.46.004774.}*

Thank you, this has been corrected.

*{\em Figure 3: Plotting column averaging kernels on a linear pressure grid is helpful for the total column, which is weighted by mass.}*

Plot has been changed

*{\em Table 2: Please organize this table by the Figure 6 sectors.}*

Table 2 has been changed, as suggested

*{\em Figure 6: Could you please add an extra bar for the mini-LHR+TCCON inversions?}*

Figure 6 has been changed, as we now show the mini-LHR+NOAA insitu sites, as suggested by the reviewers.

**\section*{Reviewer #2}**

*{\em 1. My first main concern is that the OSSEs do not account for spatially and temporally varying systematic errors in the synthetic data. This would have been an acceptable OSSE study a decade ago, but experience with GOSAT and OCO-2 data has shown that systematic errors are the main challenge with using XCO2 data for flux inversions. Indeed, this was noted by the authors on Page 2, where they stated that for GOSAT and OCO-2 "poorly characterized systematic errors compromise the accuracy of their data (Wunch et al., 2017) and limit the utility of such datasets for inferring surface flux distributions (Basu et al., 2013)." In light of this, I don't see how the authors can neglect systematics errors in their OSSEs. I am sure that the authors are aware of the numerous published OSSEs that were conducted before the launch of GOSAT and OCO that argued that future satellite observations of CO2 will significantly reduce flux uncertainties. Unfortunately, many of those OSSEs did not realistically look at the impact of systematic errors on the flux inversions. The authors must address this in their OSSEs before this study can be considered acceptable for publication.}*

We appreciate this comment, but the focus of the paper is to determine the relative importance of mini-LHR against the TCCON instruments. Both sets of synthetic data are treated the same so their ability to determine regional CO2 fluxes can be compared. Poorly characterized systematic errors *are* progressively less of an issue in the interpretation of GOSAT and OCO-2 XCO2 data but they do remain. The newest versions of OCO-2 data have made great strides in minimizing systematic errors. With respect to this reviewer, we are unsure what this study would gain from including a description of systematic errors and then applying a bias correction, without any justification for the size and nature of the systematic error.

*{\em 2. My second main concern is with the use of TCCON as a benchmark for the OSSEs. It has been shown that TCCON can provide useful information on the carbon cycle, but the network was designed mainly for satellite validation. If the focus of this manuscript is on the "potential improvements in global carbon flux estimates" associated with the mini-LHR network, the issue should be examined in the context of the added value of the mini-LHR network given*

*the exiting in situ and satellite observing systems that provide observations used in flux inversions. TCCON data are rarely used for flux inversions. At a minimum, the authors should have included the in situ surface network (the flask and quasi-continuous sites) in their OSSEs. However, I would prefer to see a comparison involving the in situ network and OCO-2 with the mini-LHR network.}*

Point well taken. As the reviewer will be aware, the ground-based networks were designed to sample large spatial scales and long temporal scales, and have an uneven distribution mainly focused on northern hemispheric maritime regions. We have now included calculations that describe the relative performance of the ground-based *in situ* network, as suggested. We fully agree (and acknowledge in the paper) that multiple data streams are necessary to infer surface fluxes at various temporal and spatial scales. Our proposed LHR network is designed primarily to help validate satellite measurements, but of course will be able to contribute to existing and future measurement networks.

*{\em 3. Another concern that I have with the OSSEs here is that the same model is used for the nature run as for the assimilation. A challenge with CO2 flux inversions is that we don't unknown how model errors will be manifested in the estimated fluxes given the information content of the data. Using the same model to produce the synthetic data and for the assimilation creates an overly optimistic scenario. I would encourage the authors to use output from another model, using different meteorological fields, to produce their synthetic data. This will provide a more rigorous OSSE and is now standard OSSE practice (See Hoffman and Atlas, Future Observing System Simulation Experiments, BAMS, Vol 9, 1601-1616, 2016).}*

We did use the same atmospheric transport model for the nature and assimilation runs but we used independent biospheric fluxes from the ORCHIDEE and CASA land biosphere models. ORCHIDEE and CASA CO2 fluxes are very different in seasonal magnitude and distribution, so we believe this is a good test for the simulated data. Our calculations are not focused on understanding the influence of model atmospheric transport error but are intended to assess the relative performance of a network of mini-LHR instruments and the current TCCON. We have added a caveat in the methods (page 8) and conclusion about results being a conservative approach. Please note we have intentionally focused on the unitless gamma metric, associated with the relative improvement in CO2 flux uncertainty, rather than only report absolute values that could be misconstrued, considering their dependence on the assumed true flux (or more precisely, on the assumed differences between the 'true' and the a priori).

*{\em 4. The authors acknowledge that cloud cover is an issue, i.e. data are collected "throughout the day during sunlight hours when clouds are not present." How was that accounted for in the OSSEs? Did they use the MERRA cloud fields to simulate data loss due to cloud cover? It is unclear if this was done. Capturing this well is important for contrasting the regional improvements in the flux estimates as data loss will be worse in some regions than others, and will vary seasonally.}*

Clear-sky measurements are determined by randomly sampling cloud coverage taken from ECMWF ERA-interim reanalysis. We agree that there are a lot of issues associated with such a simple approach to determine complicated cloud contamination probability, due to, for example, coarse spatial and temporal resolution of model cloud coverage relative to instrument view spots.

\subsection{Minor comments}

{\em 1. Page 2, line 13: define mini-LHR.}

This text has been added.

{\em 2. Page 3, line 9: it should be "result in a" and "will be the".}

This has been corrected.

{\em 3. Page 4, lines 19 and 20: Is it MERRA or MERRA-2?}

It is MERRA-2. The text has been corrected.

{\em 4. Page 4, line 28: Some words are missing here: "to get calculate spectra".}

This has been corrected.

{\em 5. Page 6, line 24: No, this is not a rigorous test of the data. Please see main Comments 1, 3, and 4 above.}

Please see above responses.

{\em 6. Page 7, line 18: How reasonable is this 5% error? On what it is based? The assumed a priori error will influence the estimated DOF. An overestimate of the a priori error will result in artificially large DOFs.}

5% of $CO_2$ will be 20 ppm, which is larger than typical discrepancies between models and data.

{\em 7. Page 8, line 3: Instrument biases really must be included in the OSSEs (see main comment).}

Please see our responses above.

{\em 8. Page 9, lines 2-4: What is the reference for this statement? It is not clear to me from what was presented that the performance shown here rivals the in situ network over North America and outperforms it in the tropics. This is why I would like to see the in situ data included in the OSSEs (see main Comment 2).}

We have now included the relative performance of the in situ network.

**\section*{Frank Hase}**

{\em The model section of this study provides useful information concerning how our capabilities of estimating CO2 fluxes could be improved by further extending our observational capabilities. The problem I have with this paper is due to the fact that it does not perform this examination on the grounds of a hypothetical network of selectable quality, but explicitly refers to a network comprised of a certain type of existing device (mini- LHR instruments). If such a reference is chosen, it seems essential to me to include a reliable characterization of the actual performance achieved by this kind of device. The noise error of a single measurement recorded with a single unit of a network is not at all the critical issue when investigating what gain could be induced by adding such a network. The impact of systematic errors (between units: site-to-site biases, and correlated errors for a selected unit: drifts) is in my opinion not adequately addressed in the paper. A proper characterization of the error budget seems an essential prerequisite to this study as the authors explicitly refer to a certain kind of existing instrumentation (similar requirements would result from the claim that a mini-LHR network would be useful for satellite validation).}

This point is well taken. As this reviewer will appreciate long-term drift and other sources of systematic error are generally difficult to assess within a global network. We have assumed zero systematic error for each of the studied measurement networks studied but acknowledge they do exist at some level albeit smaller for the more established, better characterized networks. Making this assumption allows us to compare the ability of complementary network to quantify regional CO2 fluxes. To address this reviewer comment, we have added a paragraph that outlines this point on page 7 and on page 11.

{\em Therefore, the announced instrumental study should be published before the submitted paper in order to provide a reference. A long-term side-by-side comparison of one unit with a TCCON spectrometer (spanning at least one annual cycle) would be a minimum requirement. (A previous version of the manuscript handled under AMTD manuscript number 2017-368 provided slightly more information and presented actual mini-LHR spectra, so this current version of the manuscript seems to me a further degradation of the previous presentation.)}

This version of the paper does include spectra and information about the mini-LHR (Figure 3), included as a response to comments in the review process. Realistically, we cannot include a long-term side-by-side comparison of the mini-LHR TCCON in a time fashion. In any case, having that additional information would not substantively improve what is essentially a model-led paper. The main message of the paper, which this reviewer will appreciate, is that ground-based remote sensing of CO2 (whether that is achieved using LHR or FTIR technology) can and will play an important role in understanding the global carbon cycle: directly by providing constraints on atmospheric columns and indirectly by providing ground-truth anchor points for satellite instruments.

{\em Moreover, the impression that no adequate investigation at all is attempted by the authors for establishing a reliable error budget is further consolidated by the fact that even a simple estimation of the total error budget based on plausible assumptions concerning instrument performance and a-priori knowledge of the atmospheric state is not provided. Note that TCCON uses the co-observed column of molecular oxygen for generating column-averaged dry-air mole fractions. This step is useful not only for reducing the error propagation of instrumental imperfections, but also to reduce other detrimental impacts, as e.g. errors in the assumed atmospheric temperature and ground pressure - I would therefore expect a less favourable error budget for the mini-LHR instrument.}

This reviewer will appreciate the modelling study reported here builds on studies that have already been published that include an instrument design paper (Wilson et al, 2013; doi: 10.1007/s00340-013-5531-1), a detailed error analysis (Clarke et al, 2014; doi:10.1088/0957-0233/25/5/055204), a description of autonomous field measurements (Melroy et al, 2015, doi: 10.1007/s00340-015-6172-3). Collectively, these studies provide some background on the technology and previous deployments of the instrument. These papers are cited in the current study.

{\em In summary, I would require to profoundly strengthen the part on instrument characterization and overall error budget of the proposed network (or publish these aspects before the presented kind of study), and to apply the resulting correlated measurement errors (drifts, airmass-dependent effects, site-to-site biases, etc.) for achieving a realistic estimate of the potential improvements. I do not recommend a publication of this work in its current shape.}

Respectively, we disagree with this viewpoint. We have responded above to the individual comments.